# Nova proteins direct synaptic integration of somatostatin interneurons through activity-dependent alternative splicing

Leena Ali Ibrahim[1,2,3]*[†], Brie Wamsley[4][†][‡], Norah Alghamdi[2], Nusrath Yusuf[1,3,4][§], Elaine Sevier[1,3], Ariel Hairston[1], Mia Sherer[1,3], Xavier Hubert Jaglin[4], Qing Xu[5], Lihua Guo[5], Alireza Khodadadi-Jamayran[6], Emilia Favuzzi[1,3], Yuan Yuan[7], Jordane Dimidschstein[3], Robert B Darnell[7], Gordon Fishell[1,3]*

[1]Department of Neurobiology, Harvard Medical School, Boston, United States; [2]Biological and Environmental Sciences and Engineering Division (BESE), King Abdullah University of Science and Technology (KAUST), Thuwal, Saudi Arabia; [3]Stanley Center at the Broad, Cambridge, United States; [4]NYU Neuroscience Institute and the Department of Neuroscience and Physiology, Smilow Research Center, New York University School of Medicine, New York, United States; [5]Center for Genomics & Systems Biology, New York University, Abu Dhabi, United Arab Emirates; [6]Genome Technology Center, Applied Bioinformatics Laboratories, NYU Langone Medical Center, New York, United States; [7]Laboratory of Molecular Neuro-Oncology, The Rockefeller University, New York, United States

*For correspondence:
leena.ibrahim@kaust.edu.sa (LAI);
gordon_fishell@hms.harvard.edu (GF)

[†]These authors contributed equally to this work

Present address: [‡]Department of Neurology, David Geffen School of Medicine, UCLA, Los Angeles, United States; [§]Rutgers University, New Brunswick, United States

**Abstract** Somatostatin interneurons are the earliest born population of cortical inhibitory cells. They are crucial to support normal brain development and function; however, the mechanisms underlying their integration into nascent cortical circuitry are not well understood. In this study, we begin by demonstrating that the maturation of somatostatin interneurons in mouse somatosensory cortex is activity dependent. We then investigated the relationship between activity, alternative splicing, and synapse formation within this population. Specifically, we discovered that the Nova family of RNA-binding proteins are activity-dependent and are essential for the maturation of somatostatin interneurons, as well as their afferent and efferent connectivity. Within this population, Nova2 preferentially mediates the alternative splicing of genes required for axonal formation and synaptic function independently from its effect on gene expression. Hence, our work demonstrates that the Nova family of proteins through alternative splicing are centrally involved in coupling developmental neuronal activity to cortical circuit formation.

## Editor's evaluation

This is an important study that explores the roles of a set of RNA binding proteins on the connectivity and development of a prominent class of neocortical interneurons. The authors provide convincing evidence that these proteins regulate alternative splicing of key effector genes in these neurons and regulate neuronal inputs and outputs in an activity-dependent manner.

## Introduction

Somatostatin cortical interneurons (SST cINs) constitute ~30% of all inhibitory interneurons in the cerebral cortex. They are crucial for gating the flow of the sensory, motor, and executive information

necessary for the proper function of the mature cortex (*Fishell and Rudy, 2011*; *Kepecs and Fishell, 2014*; *Tremblay et al., 2016*). In particular, Martinotti SST cINs, the most prevalent SST cIN subtype, are present in both the infragranular and supragranular layers of the cortex and extend their axons into Layer 1 (L1; *Rudy et al., 2011*; *Lim et al., 2018*; *Ascoli et al., 2008*; *Nigro et al., 2018*; *Pouchelon et al., 2021*). They specifically target the distal dendrites of neighboring excitatory neurons, thus providing the feedback inhibition necessary for modulating dendritic integration (*Adler et al., 2019*; *Kapfer et al., 2007*; *Silberberg and Markram, 2007*). These roles are dependent upon the ability of SST cINs to form specific synaptic connections with select excitatory and inhibitory cell types during development (*Favuzzi et al., 2019*).

The mechanisms responsible for generating the precise functional connectivity of SST cINs are poorly understood. Early neuronal activity has emerged as an important factor in directing the maturation of cINs (*Wamsley and Fishell, 2017*). In addition, recent work has implicated activity as being centrally involved in alternative splicing (*Eom et al., 2013*; *Furlanis and Scheiffele, 2018*; *Iijima et al., 2011*; *Lee et al., 2007*; *Lee et al., 2009*; *Mauger et al., 2016*; *Quesnel-Vallières et al., 2016*; *Vuong et al., 2016*; *Vuong et al., 2018*; *Xie and Black, 2001*). However, whether these processes are coupled within SST cINs has not been explored.

The Nova family of RNA-binding proteins (Nova1 and Nova2) have been shown to control the splicing and stability of transcripts encoding a variety of neurotransmitter receptors, ion channels, and transmembrane cell adhesion molecules known to affect synaptogenesis and excitability (*Dredge and Darnell, 2003*; *Eom et al., 2013*; *Saito et al., 2016*; *Saito et al., 2019*; *Ule et al., 2005*; *Ule et al., 2006*; *Yano et al., 2010*). Notably both Nova1 and Nova2 are strongly expressed within cINs during periods of synaptogenesis and as such represent promising effectors that may direct the maturation of SST cINs.

Here, we report that neuronal activity strongly influences efferent SST cIN connectivity. We show that the conditional loss of *Nova1* or *Nova2* phenocopies the effect of dampening activity during circuit assembly, leading to a loss of their efferent inhibitory output. At a molecular level these changes are mediated by a Nova-dependent program, which controls gene expression and alternative splicing of mRNAs encoding for pre- and post-synaptic proteins. Demonstrating a direct link between activity, Nova function, and inhibitory output, increasing activity using NachBac in Nova2 knockouts fails to enhance SST inhibitory output. Conversely, overexpression of *Nova2* within SST cINs marginally increases SST inhibitory output, a phenotype that can be suppressed by damping neuronal activity within these cells. Thus, our work indicates that early activity through a Nova-dependent mechanism is required for the proper establishment of SST cIN connectivity and maturation.

## Results

### Neuronal activity affects the synaptic development of SST cINs

The cortex exhibits a variety of dynamic network activity patterns during cortical synaptogenesis (*Allene and Cossart, 2010*; *Garaschuk et al., 2000*; *Yang et al., 2009*). These are comprised by both spontaneous and sensory evoked events (*Garaschuk et al., 2000*; *Minlebaev et al., 2011*; *Yang et al., 2013*; *Pouchelon et al., 2021*; *Ibrahim et al., 2021*). While inhibitory cortical interneurons (cINs) are recruited by these activities (*Cossart, 2011*; *Le Magueresse and Monyer, 2013*), whether this influences somatostatin (SST) cIN development has not been fully established. To address the impact of activity on these cINs, we chose to selectively and cell-autonomously dampen or augment their excitability during the first few weeks of development. This represents a perinatal period in cIN development during nascent circuit formation, where they are robustly forming or losing synaptic contacts (*Allène et al., 2008*; *Minlebaev et al., 2011*; *Yang et al., 2013*; *Yang et al., 2009*). SST cINs in the primary somatosensory cortex (S1) were targeted using AAV viral injections in *Sst^Cre* mice crossed with a conditional synaptophysin1-eGFP (Syp-eGFP) mouse, which functions as a presynaptic reporter (*Rosa26^{LSL-tTa}*;Tg-TRE::Syp-eGFP, *Figure 1A*; *Basaldella et al., 2015*; *Li et al., 2010*; *Wamsley et al., 2018*). To modulate the activity of SST cINs, these mice were injected at P0 with Cre-dependent AAVs that drive the expression of either KIR2.1 (AAV-Syn-DIO-KIR2.1-P2A-mCherry) or NaChBac (AAV-Syn-DIO-NaChBac-P2A-mCherry) channels coupled to mCherry reporter (*Figure 1A*). Both channels are voltage-sensitive and have proven to be useful tools to manipulate cellular excitability. The KIR2.1 channel is an inward rectifying potassium channel, which upon overexpression

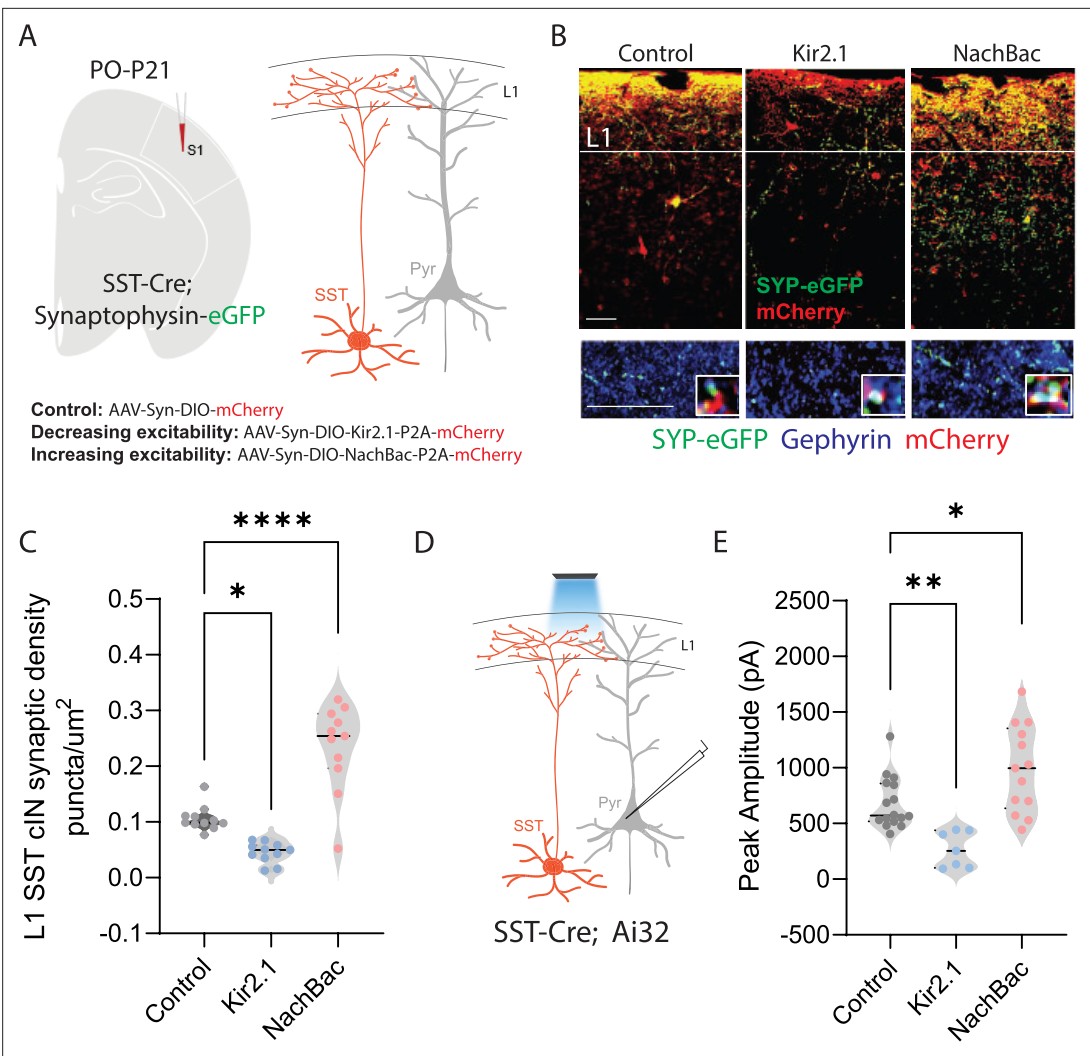

**Figure 1.** Neuronal activity affects the synaptic development of SST cINs. (**A**) Schematic ( (**A**) has been adapted from the Research Article Summary Schematic from *Bernard et al., 2022*) of genetic alleles (left) and experimental approach (middle), *Sst^Cre;Syp-eGFP* pups were injected with a conditional virus; either *AAV2/1-Flex-Kir2.1-P2A-mCherry, AAV2/1-Flex-NaChBac-P2A-mCherry,* or Control *AAV2/1-Flex-mCherry* within the S1 cortex at postnatal day 0 (**P0**). Schematic of the efferent connectivity of SST +Martinotti cells (right). (**B**) Upper panels: Immunostaining (IHC) of *Sst^Cre;Syp-eGFP* in layer 1 (**L1**) of S1 cortex at P21 showing Syp-eGFP (green, anti-GFP) and axons (red, anti-RFP) from *control, Kir2.1, or NaChBac* injected SST +Marintonotti cINs (scale bar 50 um). Lower panels: visualization of Syp-eGFP (green, anti-GFP) and Gephyrin +puncta (blue, anti-Gephyrin) in L1 SST +cINs (scale bar 20 um). Inset shows a higher magnification image of the puncta overap (Red, mCherry axons). (**C**) Quantification of synaptic puncta (RFP+/GFP+/Gephyrin +overlap) of control, Kir2.1, and NaChBac expressing SST +cINs within L1 (n=3–4 mice each, 9 sections each; pVal**=0.008, ***=0.0001). (**D**) Left, schematic of optogenetic activation of SST neurons using *SST^Cre*::Ai32 mice injected with either *AAV2/1-Flex-Kir2.1-P2A-mCherry, AAV2/1-Flex-NaChBac-P2A-mCherry,* or Control *AAV2/1-Flex-mCherry* within the S1 cortex at postnatal day 0 (**P0**) and recording from Pyramidal neurons (clamped at 0 mV) in Layer 5 of Primary Somatosensory cortex (**S1**) at P21. (**E**) Quantification of SST output onto Pyramidal neurons, Peak amplitude of the Inhibitory post synaptic current (IPSC) (Control Peak Amplitude: 663.47±18.7 pA, Kir2.1: 185±19.78 pA, NaChBac: 927.7±28.5 pA).

lowers the resting membrane potential towards the reversal potential of $K^+$ (~90 mV) (*Bortone and Polleux, 2009*; *De Marco García et al., 2011*; *Karayannis et al., 2012*; *Priya et al., 2018*; *Yu et al., 2004*), thus reducing neuronal excitability. The NaChBac channel has an activation threshold that is 15 mV more negative than endogenous voltage-gated $Na^+$ channels and remains open for 10 times longer (*Lin et al., 2010*) and therefore augments excitability.

To assess the development of the synaptic efferents of infected SST cINs, we allowed pups to mature until juvenile age. The somatosensory cortex was then subjected to immunohistochemistry (IHC) to visualize pre-synaptic (SST + cIN-mCherry+-Syp-eGFP+) compartments and post synaptic components and subjected to puncta analysis (*Figure 1B*). We quantified SST efferent synapses

identified through the colocalization of the virally mediated mCherry reporter, Syp-eGFP and the postsynaptic marker gephyrin (mCherry+/GFP+/gephyrin +puncta), as a proxy for synaptic contacts (*Ippolito and Eroglu, 2010*). In SST cINs, KIR2.1 expression resulted in a significant reduction of L1 SST cIN efferent synaptic puncta in comparison to control cells ($0.109\pm0.009$ puncta/μm² CTL vs $0.049\pm0.006$ puncta/um² KIR2.1; *Figure 1C*). By contrast, the overexpression of NaChBac within SST cINs resulted in a robust increase in L1 synaptic puncta ($0.109\pm0.009$ puncta/um² ctl vs $0.218\pm0.030$ puncta/um² NaChBac; *Figure 1C*). Additionally, when we optogenetically activated SST neurons using a conditional channelrhodopsin mouse line *Rosa26$^{LSL-hChR2}$* (Ai32) and recorded from pyramidal cells (*Figure 1D*), the inhibitory output was also affected. Kir2.1 expression resulted in a significant reduction in the output of SST + cINs onto pyramidal cells compared to controls ($680\pm14$ pA ctl vs $266.5\pm21$ pA Kir2.1, *Figure 1E*). By contrast, the overexpression of NachBac resulted in an increase in the inhibitory output of these cells ($680\pm14$ pA ctl vs $989.33\pm28$ pA, *Figure 1E*). These results suggest that activity has a profound effect on the density of SST cIN axons, synapses and inhibitory output. Dampening excitability decreases the number of efferent synaptic structures and axonal arbors of SST cINs, while augmenting it increases both.

## Neuronal activity influences alternative splicing and Nova expression within SST cINs

A growing number of studies indicate that activity-dependent alterative splicing (AS) contributes to the regulation of gene expression and the fine-tuning of transcriptional programs related to synaptic refinement (*Eom et al., 2013*; *Iijima et al., 2011*; *Fuccillo et al., 2015*, *Mauger et al., 2016*; *Quesnel-Vallières et al., 2016*; *Vuong et al., 2016*). This prompted us to test whether neuronal activity itself changes the level of AS within SST cINs during circuit formation, independent of the changes in gene expression. To do so, we used electro-convulsive shock (ECS) during peak synaptogenesis (P8) in mice with genetically labeled SST cINs (*Sst$^{Cre}$*; *Rosa26$^{LSL-tdTomato}$* (Ai9)). The ECS method generates an acute and reproducible increase in neuronal activity in vivo (*Guo et al., 2011*; *Ma et al., 2009*), resulting in increased expression of immediate early genes (IEG) such as *Fos*, *Egr1*, *Npas4,* and *Arc* (*Figure 2— figure supplement 1A–F*), analogous to that observed with KCl treatment in vitro but with the added advantages of being in vivo and transient. Two to 3 hr following ECS, we isolated SST cINs from the S1 cortex of *Sst$^{Cre}$*; *Rosa26$^{LSL-tdTomato}$* (Ai9) animals using fluorescence-activated cell sorting (FACS; *Figure 2A* and *Figure 2—figure supplement 1A*). Sorted SST cINs were used to prepare cDNA libraries that were subsequently sequenced in order to investigate changes in AS (spliced exon: SE, mutually exclusive exons: MXE, retained intron: RI, alternative 5′ splice site: A5, alternative 3′ splice site: A3; *Figure 2A* right). We found 312 transcripts differentially spliced between sham/control and ECS (FDR <0.05, $|\Delta\psi|\geq0.1$ threshold), comprised by 139 SE events (57 excluded and 82 included exons), 66 RI events (13 excluded introns and 53 included), 31 MXE events (29 excluded and 26 included exons), 13 A5 events (1 excluded and 12 included exons), and 39 A3 (23 excluded and 16 included exons) (*Figure 2B*).

Utilizing the SST cIN transcriptome as a reference, we performed gene ontology (GO) analysis to ask if the genes subject to alternative splicing (AS) were enriched for specific functional categories within these neurons. GO analysis of the genes that underwent activity dependent AS belong to specific ontological categories, such as synapse maturation, synaptic transmission, and axonal growth (*Figure 2C* and *Figure 2—figure supplement 2B*). In addition, and as expected we also observed activity-dependent changes in gene expression (*Figure 2D*). However, the overlap between the genes subjected to AS vs GE was only ~1.2% of all differentially expressed genes (*Figure 1E*). When we compared the overlapped genes (those that underwent both GE and AS changes), we observed that for many synaptic genes, the level of AS changes was higher than the changes of the same genes at the transcript level (*Figure 1F*). This suggests that the changes observed in synaptic genes for AS are independent of their changes in transcription level. For example, we observed and validated (*Supplementary file 2b*) that within activity-stimulated SST cINs the *Nrxn1* mRNAs exclude exon 10. Notably, this exon lies within a laminin-protein coding domain important for the cell adhesion properties of Nrxn1 at the synapse (*Figure 2G*, control, grey vs ECS, green). In contrast, while we also observed GE changes in Nrxn1, the level of AS change was higher.

We next asked whether the activity-dependent AS genes formed a protein-protein interacting network (PPI) based on previously established direct protein interactions in vivo (*Rossin et al., 2011*).

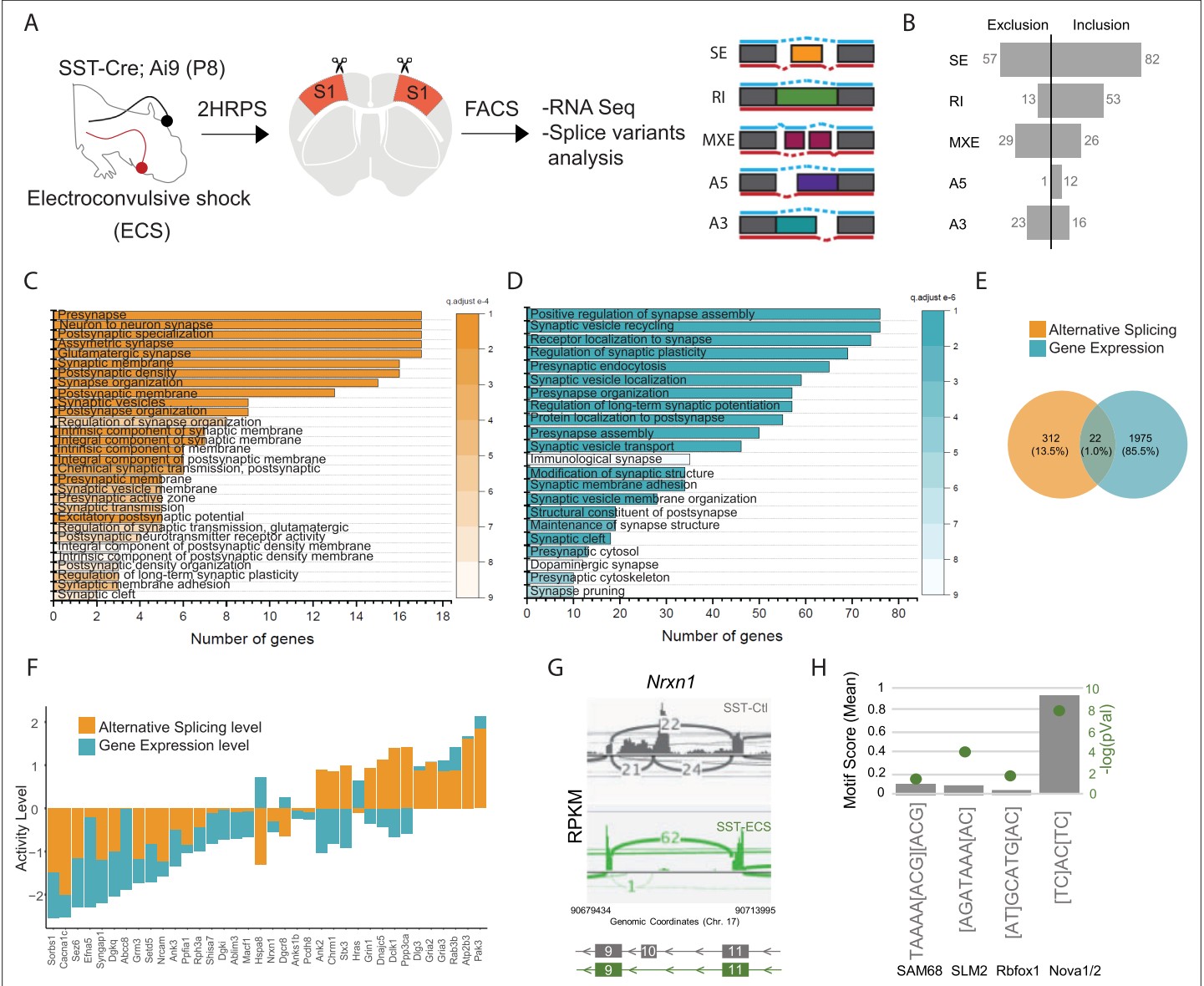

**Figure 2.** Neuronal activity influences alternative splicing and Nova expression within SST cINs. (**A**) Schematic of experimental approach: Postnatal day 8 (**P8**) *Sst^Cre^; Rosa26^LSL-tdTomato^* (Ai9) pups were subjected to electroconvulsive shock (ECS) (left). Following 2–3 hours the S1 cortex was isolated and SST + cINs were FACS purified (middle). SST + cINs were then prepared for RNAseq to assess changes in gene expression and alternative splicing. Splicing changes are divided into the major alternative structural motifs: single exon, SE, retained intron, RI, mutually exclusive exons, MXE, alternative 5' splice site, A5, alternative 3' splice site, A3 (right). (**B**) Histogram of the magnitude of activity-dependent splicing changes within SST + cINs subjected to ECS compared to sham SST + cINs (FDR <0.5, fold <0.1> ), depicting 139 differential spliced SE (82 SE included, 57 SE excluded), 66 differential spliced RI (53 RI included, 13 RI excluded), 55 differential spliced MXE (26 included, 29 excluded), 13 differential spliced A5 (12 included, 1 excluded), 39 differential spliced A3 (16 included, 23 excluded). (**C**) Gene Ontology (GO) analysis of differentially alternatively spliced (AS) genes (Orange color) under synaptic categories. (**D**) Gene Ontology (GO) analysis of differentially expressed genes (GE) (Teal color) under synaptic categories. (**E**) Overlap of all differentially expressed (teal) and differentially spliced (orange) genes under ECS vs control conditions. Below: overlap for synaptic gene only. (**F**) Comparison of activity level of the overlapped synaptic genes (genes that have both AS and GE changes). Activity level is calculated by considering both FC and pvalue. (**G**) Sashimi plot illustrating *Nxrn1* exon 10 exclusion in activity-induced SST cINs in green (bottom) compared to sham SST cINs in grey (top). Reads per kilobase of transcripts (RPKM) gives the count of the number of transcripts for a specfic isoform. (**H**) Histogram of the average motif enrichment score of known activity-regulated splicing factors KHDRBS1 (Sam68), KHDRBS2 (SLM2), Rbfox1 and Nova1/2 (right). Green dots represent -log10 adjusted p value (right Y-axis) for motif enrichment scores, only significant enrichment shown.

The online version of this article includes the following figure supplement(s) for figure 2:

**Figure supplement 1.** Acute increases in neuronal activity induces immediate early gene expression and differential splicing within SST + cINs in vivo.

**Figure supplement 2.** Neuronal activity influences alternative splicing and Nova expression within SST cINs.

Notably, the genes subjected to activity-dependent AS within SST cINs form highly connected networks illustrating they likely function together to support pre-synaptic vesicle function (*Figure 2—figure supplement 2C*, pink), post-synaptic organization and receptor-associated synaptic components (*Figure 2—figure supplement 2C*, blue; pVal <0.0009, 1000 permutations). These genes among others include: *Hspa8, Nrxn1, Syngap1, Cacna1c, Ppp3ca,* and *Grin1* (*Figure 2—figure supplement 2C*). These results indicate that augmenting activity within SST cINs during nascent circuit development robustly increases AS events and most of the spliced mRNAs are genes specifically related to axonal development and synaptic transmission (*Figure 2—figure supplement 2C*, pink and blue, respectively).

We next sought to identify RNA-binding proteins (RNABPs) that could mediate activity-dependent AS events within SST cINs. To do so, we utilized the RNAseq experiments described above (control vs. ECS) to perform a motif enrichment analysis that utilizes position probability matrices of binding motifs from 102 RNABPs (e.g. PTBP1/2, FUS, ELAVL4, SRRM4, Rbfox1, FMR1, Nova1, Nova2) (*Liu et al., 2017*; *Park et al., 2016*; *Yang et al., 2016*). Previous HITS-CLIP analysis has revealed that Nova1 and Nova2 share an almost identical RNA-binding domain (YCAY) (*Licatalosi et al., 2008*; *Ule et al., 2006*; *Yuan et al., 2018*). Strikingly, the Nova-binding motif was found to be significantly enriched within activity-dependent targets and at a higher frequency than other neuronal splice factors (e.g. Sam68 (KHDRBS1), SLM2 (KHDRBS2), and Rbfox1) (pVal <0.0001; *Figure 2H*). This finding implicates Nova proteins as playing a fundamental role in directing SST cIN activity-dependent AS.

## Neuronal activity during cortical development influences the expression and localization of Nova proteins in SST cINs

We next examined the expression of Nova1 and Nova2 within SST cINs across development and whether their expression is affected by changes in neuronal activity. Utilizing IHC and genetic fate mapping, we observed that the expression of the Nova family (Nova1 and Nova2) proteins begins within cIN populations soon after they become postmitotic and expressed in 100% of SST and PV cINs by adulthood (*Figure 3—figure supplement 1A–B*). For comparison, we also examined Nova expression in 5HT3aR cINs (*Figure 3—figure supplement 1B*) within this same region. To specifically examine the expression of Nova1 and Nova2 during SST cIN synaptogenesis, we performed quantitative-PCR (qPCR) on FACS isolated cINs from the S1 cortex of Tg-Lhx6::eGFP mice at P2, P8, and P15. The Tg-Lhx6::eGFP mice express eGFP in both SST and Parvalbumin (PV) cINs (medial ganglionic eminence derived cINs) soon after they become postmitotic. We found that both *Nova1* and *Nova2* are expressed within all SST and PV cINs across the first two weeks of postnatal development, coinciding with nascent circuit development (*Figure 3—figure supplement 1C*). Taken together, we find that both Nova1 and Nova2 proteins are highly expressed in all SST cINs during circuit integration and may therefore control integral aspects of their development through activity-dependent alternative splicing.

We next investigated whether Nova1 and Nova2 are activity-regulated within SST cINs by examining both their expression and localization during the peak of nascent circuit integration. To do so, we subjected *Sst^Cre*; *Rosa26^LSL-tdTomato* (Ai9) mice to ECS during synaptogenesis, similar to what was done in *Figure 2*. Next, we performed both RNA Sequencing and qPCR for *Nova1* and *Nova2* within FAC sorted cINs from S1 cortex of either control or ECS-treated animals. Following 2 hr post seizure-induction (2 HRPS), we found that the mRNA expression levels of both *Nova1* and *Nova2* were increased in ECS-treated SST cINs compared to controls (*Nova1*: 28.4±5.59 ECS vs 7.61±0.25 control; *Nova2* pVal = 0.002: 10.32±1.80 ECS vs 6.59±0.28 control pVal = 0.005, *Figure 3—figure supplement 1E* and *Figure 3A*). Next, we probed Nova1 and Nova2 protein levels using western blot (WB) of sorted SST cINs from S1 cortex. Consistent with an activity-mediated upregulation in Nova expression, we found a significant increase in both Nova1 and Nova2 protein levels (0.824±0.0412 pixel density (pd) Nova1 control vs 5.62±0.969 pd Nova1 2HRPS, pVal = 0.038 and 0.997±0.409 pd Nova2 control vs 5.7±0.582 pixel density Nova2 2HRPS, pVal = 0.022, pixel densities normalized to ß-Actin; *Figure 3B*). Thus, these results confirm an increase in Nova mRNA and Nova protein expression in SST cINs following an acute increase in neuronal activity.

Following seizure activity Nova proteins have been shown to translocate into the nucleus within excitatory neurons (*Eom et al., 2013*). We next sought to explore whether manipulating activity also influences intracellular localization of Nova proteins within SST cINs (*Figure 3C*). We hypothesized that

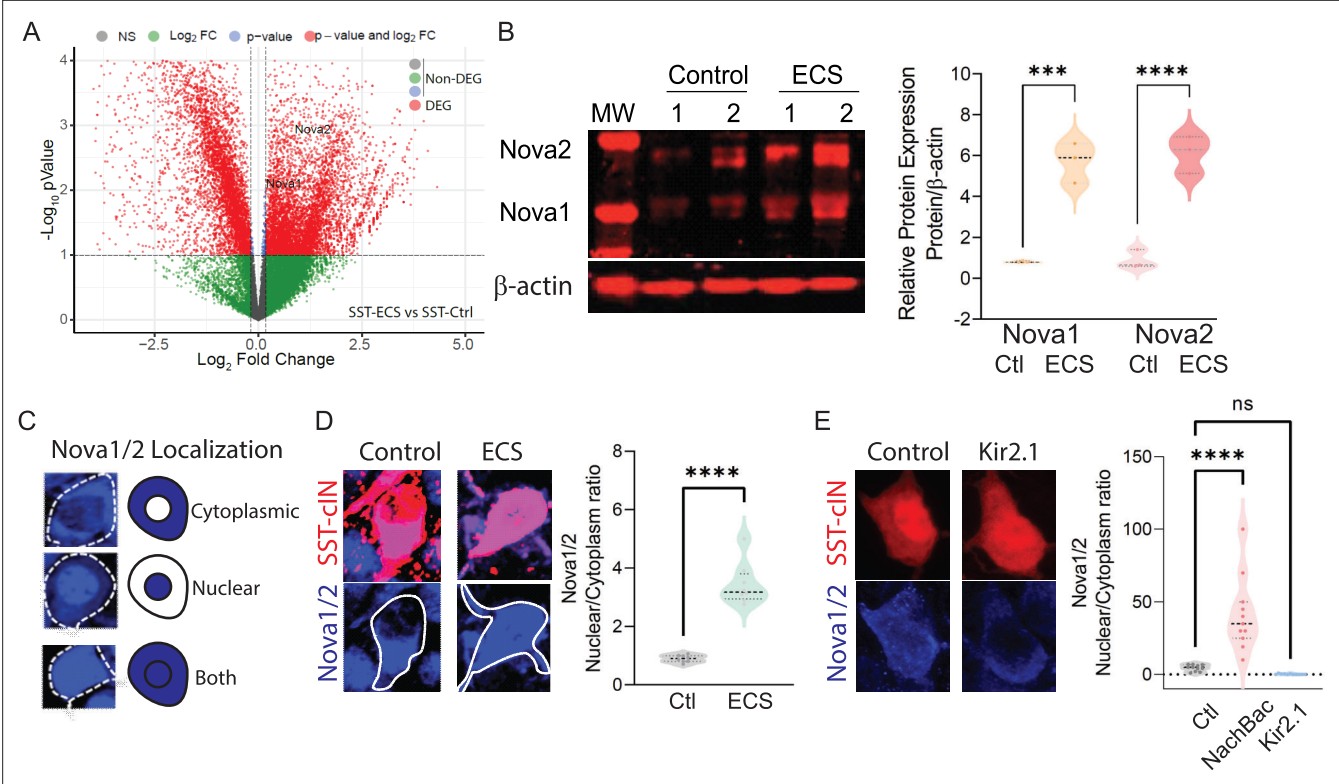

**Figure 3.** Neuronal activity during cortical development influences the expression and localization of Nova proteins in SST cINs. (**A**) Volcano plot of RNA seq data showing Nova1 and Nova2 upregulation in SST-cINs in ECS vs control. (**B**) Upper panel, western blot showing Nova1 and Nova2 protein expression in control (lanes 2 and 3) versus ECS induced SST cINs (lanes 4 and 5). Lower panel, same western blot showing expression of b-actin across lanes. Right, Quantification of the western blot data. Nova1 and Nova2 protein expression relative to β-actin in control versus ECS induced SST cINs (n=3 mice, S1 cortex only; *pVal = 0.038, Nova1; *pVal = 0.022, Nova2; Source Data not available due to loss of data file during lab move). (**C**) Representative scoring criteria for Nova1/2 localization within SST cINs: IHC of Nova1/2 (blue, anti-Nova1/2) in selective SST + cINS exemplifying the Nova1/2 expression in: cytoplasm only (top), nucleus only (middle) and in both cytoplasm and nucleus (bottom). (**D**) Left, representative images of Nova1/2 expression (blue) in SST cINs (red) under normal versus ECS. Right, Quantification of the ratio of nuclear to cytoplasmic localization of Nova1/2 in SST + cINs of control animals (grey) and ECS animals (green) (n=3 mice, S1 cortex; **pVal = 0.001). (**E**) Left, representative images of Nova1/2 expression (blue) in SST cINs (red) using control mCherry versus Kir2.1-mCherry virus injection. Right, Quantification of the ratio of nuclear to cytoplasmic localization of Nova1/2 in SST + cINs of control AAV2/1-Syn-DIO-mCherry (grey) versus AAV2/1-Syn-DIO-NaChBac-P2A-mCherry (pink) versus AAV2/1-Syn-DIO-Kir2.1- P2A-mCherry (blue) injected animals. (n=11 mice, S1 cortex,~30 cells each; ***pVal = 0.0004, NachBac; ***pVal = 0.0001, KIR2.1).

The online version of this article includes the following figure supplement(s) for figure 3:

**Figure supplement 1.** Nova 1 and 2 alternative splicing factors expression within cortical interneurons (cINs).

an activity-mediated increase would direct Nova proteins to the nucleus. We therefore analyzed the ratio of Nova expression within the nucleus versus the cytoplasm of SST cINs following ECS (*Figure 3D*) or after constitutive activity-modulation across the first postnatal month (DIO: AAV injections of KIR2.1 or NaChBac into *Sst*^cre^ animals at P0; *Figure 3E*). First, we examined Nova localization using IHC 2 HRPS following ECS within the S1 cortex of P8 *Sst*^Cre^; *Rosa26*^LSL-tdTomato^ (Ai9) mice (*Figure 3C–D*). We quantified the proportions of SST cINs that express Nova proteins most prominently within the nucleus versus the cytoplasm by taking multi-Z-stack images of SST cINs and utilizing DAPI to demark the nuclear boundary. We found that Nova localization was observed in three basic patterns in SST cINs: restricted to the cytoplasm, nuclear restricted, or a combination of both nuclear and cytoplasmic expression (*Figure 3C*). At P8, in the majority of SST cINs, Nova is either restricted to the cytoplasm or expressed in both the nucleus and cytoplasm (*Figure 3D*). Following an acute increase in activity (ECS), we found a significant increase in the ratio of nuclear to cytoplasmic Nova protein within SST cINs (0.948±0.055 ratio in control versus 3.65±0.465 ratio in ECS, pVal = 0.001, *Figure 3D* right). Next, by utilizing the same analytical approach, we examined Nova protein localization in S1 of P21

mice that express either KIR2.1 or NaChBac along with a mCherry reporter (*Figure 3E*). We found a substantial increase in the ratio of SST cINs that localized Nova protein in the nucleus compared to the cytoplasm in NaChBac-expressing expressing SST cINs compared to control cells (4.75±0.678 ratio control vs 40.91±8.41 ratio NaChBac, pVal = 0.0004; *Figure 3E* right). In contrast, we found a significant decrease in the ratio of nuclear to cytoplasmic Nova protein expression within SST cINs injected with KIR2.1 (4.75±0.678 ratio control vs 0.265±0.104 ratio KIR2.1, pVal=<0.0001; *Figure 3E* right). Most strikingly, we also observed more than half of SST cINs subjected to KIR2.1 either do not express Nova or have substantially reduced levels of Nova protein expression (*Figure 3—figure supplement 1F–G*), suggesting that normal levels of activity are needed for maintaining Nova protein expression in the cell. Altogether these data indicate that during synaptogenesis Nova protein expression and localization within SST cINs is strongly modulated by acute or persistent changes in activity.

## Nova1 and Nova2 control distinct AS networks within SST cINs

To address whether Nova1 and Nova2 differentially affect connectivity and maturation, we asked what AS networks they control within SST cINs during development. Given that they share a very similar RNA-binding motif and are found associated with one another in vivo, they were thought to function cooperatively (*Licatalosi et al., 2008*; *Racca et al., 2010*; *Yuan et al., 2018*). However recently, it has been shown that in addition to their synergistic roles, Nova1 and Nova2 proteins each control distinct AS gene networks (*Saito et al., 2019*; *Saito et al., 2016*). We thus chose to examine changes in AS within SST cINs in Nova1, Nova2 or Nova1/2 compound conditional knockout (cKO) mice (*Saito et al., 2019*; *Yuan et al., 2018*). Using FAC sorting, we isolated SST cINs from *Sst^Cre^;Nova1^F/F^* or *Sst^Cre^;;Nova2^F/F^* or *Sst^Cre^;Nova1^F/F^; Nova2^F/F^* double knockout (dKO) mice on an Ai9 reporter background (referred to henceforth at *Sst-Nova1*-cKO, *Sst-Nova2*-cKO and *Sst-Nova1/2*-dKO, respectively) at P8 (*Figure 4A*). We prepared cDNA libraries from FAC sorted SST cINs, performed RNA sequencing and assessed AS changes between control SST cINs versus each of these mutant alleles. Compared to wild type controls, *Nova1* loss resulted in 124 altered AS events (81 excluded and 43 included), *Nova2* loss led to 339 altered AS events (217 excluded and 122 included) and *Nova1/2*-dKO exhibited 270 altered AS events (162 excluded and 108 included) (FDR <0.05; *Figure 4B*). Notably, within SST cINs, the loss of *Nova2* results in the largest number of changes in mRNA splicing events compared to compound loss of either *Nova1* or both *Nova* genes. Interestingly, the loss of *Nova1*, *Nova2*, or *Nova1/2*-dKO also resulted in significant gene expression changes within SST-cIN (*Figure 4—figure supplement 1A–F*). However, similar to what we observed with ECS, many of the genes that were subjected to AS (*Figure 4C*) were independent from the genes that underwent changes in gene expression (*Figure 4D*). Amongst the common genes (between AS and GE), many synaptic genes showed higher levels of AS changes compared to gene expression changes (*Figure 4D*).

We next assessed the overlap of changes in AS events observed within each mutant (*Figure 4—figure supplement 2A–C*). We found the number of alterations in *Sst-Nova2*-cKO AS events that overlap with *Sst-Nova1/2*-dKO is almost three times higher than that observed when comparing the overlap between *Sst-Nova1/2*-dKO and *Sst-Nova1*-cKO (i.e. 62 altered *Sst-Nova2*-cKO AS events coincided with the 162 observed in *Sst-Nova1/2*-dKO versus an overlap of only 25 AS events that were altered in *Sst-Nova1*-cKO mutants, *Figure 4—figure supplement 2B and C*). By contrast, less than 15% of the altered *Sst-Nova1* AS genes overlap with changes observed in *Sst-Nova2*-cKO mutants (i.e. only 28 of the 217 *Sst-Nova2*-cKO events were altered in *Sst-Nova1*-cKO mutants; *Figure 4—figure supplement 2A and B*). Interestingly, *Sst-Nova1/2*-dKO mutants exhibited less altered splicing events than the single *Sst-Nova2*-cKO mutant, suggesting that some inclusion and exclusion AS events are antagonistically directed by Nova1 and Nova2. Additionally, we performed a correlation analysis within and between *Nova1, Nova2,* and *Nova1/2*-dKO to assess whether the type of splicing events is correlated. Similar to above, we observed a higher correlation in exclusion events between *Nova2* and *Nova1/2*-dKO, compared to *Nova1* and *Nova1/2*-dKO (*Figure 4—figure supplement 2D*).

To infer their specific biological functions, we performed GO analysis on the altered AS events from each mutant and then asked whether the affected AS events form direct PPI networks. GO analysis of the *Sst-Nova1*-cKO targets did not result in any significant enrichment of specific functional categories (below an FDR of 0.05) however it did organize genes into categories such as RNA binding, ion binding, and catalytic activity (*Figure 4—figure supplement 1G*). *Sst-Nova1*-cKO AS genes formed a relatively indistinct small sparse PPI network (pVal <0.09) representing vesicle-transport

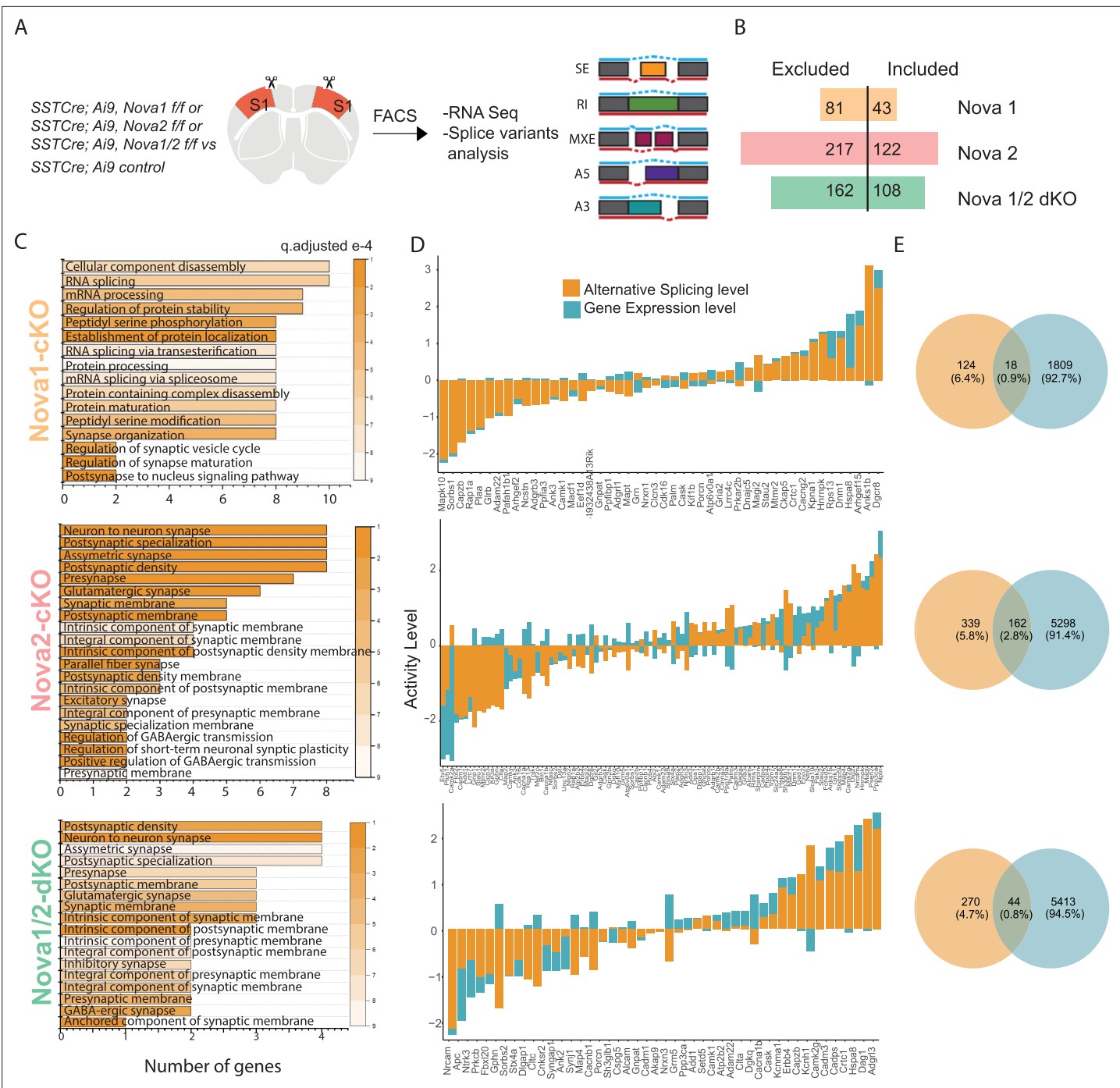

**Figure 4.** Nova1 and Nova2 control distinct alternative splicing (AS) networks within SST cINs. (**A**) Schematic demonstrating the mouselines used for FACS sorting and subsequent RNA sequencing and splice variant analysis: SST cINs from *Sst-Cre;Nova1F/F* (*Sst-Nova1*-cKO) or *Sst-Cre;Nova2F/F* (*Sst-Nova2*-cKO) or *Sst-Cre;Nova1F/F/Nova2F/F* (*Sst-Nova1/2*-dKO) mice on an Ai9 reporter background (referred to henceforth at *Sst-Nova1*-cKO, *Sst-Nova2*-cKO and *Sst-Nova1/2*-dKO, respectively). (**B**) Plot showing the number of alternative splicing (AS) events in *Sst-Nova1*-cKO, *Sst-Nova2*-cKO or *Sst-Nova1/2*-dKOs. *Nova1* loss resulted in 124 altered AS events (81 excluded and 43 included), *Nova2* loss led to 339 altered AS events (217 excluded and 122 included) and double mutants exhibited 270 altered AS events (162 excluded and 108 included; FDR <0.05). (**C**) Gene Ontology (GO) analysis of differentially alternatively spliced (AS) genes under synaptic categories for *SST-Nova1*-cKO (orange label, top panel); *Sst-Nova2*-cKO (pink label, middle panel) and for *Sst-Nova1/2*-dKO (green label, bottom panel). Color bar indicated q-adjusted values for splice variant expression. (**D**) Comparison of the level of alternative splicing activity vs gene expression for the overlapped synaptic genes (i.e., genes that show both AS and GE changes) for *Sst-Nova1*-cKO (top panel), *Sst-Nova2*-cKO (middle panel) and for *Sst-Nova1/2*-dKO (bottom panel). Activity level is calculated by considering both Fold change

*Figure 4 continued on next page*

*Figure 4 continued*

and pValue for each gene. (**E**) Percentage of genes that overlap between gene expression and alternative splicing changes FC >0.5 for *Sst-Nova1*-cKO (top), *Sst-Nova2*-cKO (middle) and *Sst-Nova1/2*-dKO (bottom panels).

The online version of this article includes the following figure supplement(s) for figure 4:

**Figure supplement 1.** Nova2 controls most of the gene expression and splicing events of the Nova1/2 family within SST + cINs and these events coalesce into GO categories and PPI networks related to pre- and post-synaptic development of SST cINs.

**Figure supplement 2.** Overlap in Alternative Splice events between SST-Nova 1, SST-Nova 2 and SST-Nova1/2 dKO.

and nucleic-acid binding pathways (*Figure 4—figure supplement 1J*, pink shaded). In contrast, *Sst-Nova2*-cKO and *Sst-Nova1/2*-dKO AS genes organized into several shared significant GO categories such as neuron projection, axon, cell-cell junction, and synaptic function (FDR <0.05) (*Figure 4—figure supplement 1H, I*). *Sst-Nova1/2*-dKO AS genes also organized into some unique categories, which were involved in postsynaptic specialization, dendrite, and synaptic vesicle membrane (FDR <0.05) (*Figure 4—figure supplement 1H*). We next asked if the AS genes affected in *Sst-Nova2*-cKO and *Sst-Nova1/2*-dKO were predicted to function together in a PPI network representing specific biological processes (*Figure 4—figure supplement 1K–L*). Perhaps not surprisingly both formed highly connected significant PPI networks (pVal <0.0009, 1000 permutations) representing multiple pathways for vesicle-transport, pre- and post-synaptic function and organization, as well as Ca$^{2+}$ signaling (*Figure 4—figure supplement 1K–L*, pink and green respectively). Interestingly, the PPI network for *Sst-Nova2*-cKO uniquely includes numerous glutamate receptors and their adaptors, respectively (e.g. Grin2b, Grik1, Gria3, Grm5 and Grip1, Sharpin, Dlg2) (*Figure 4—figure supplement 1K*, highlighted pre-synaptic genes in pink and post-synaptic genes in green). Altogether these results suggest that considering the Nova family as a whole, Nova2 (compared to Nova1) is the main driver of AS and importantly, may be most relevant for synaptic development of SST cINs.

### *Sst-Nova1 and Sst-Nova2* mutants have impaired afferent and efferent connectivity

To confirm our predictions from the AS analysis of conditional *Nova* mutants, we next sought to determine the effect of the loss of *Nova1* and *Nova2* on SST cIN synaptic development and function. To this end, we assessed the requirement for *Nova1* and/or *Nova2* for both the anatomical connectivity and physiological properties of SST cINs. *Sst-Nova1/2*-dKO mice were smaller in size and while generated at Mendelian ratios, many died as early as P8, and offspring often exhibited seizures (*Figure 5—figure supplement 1A*). In the single KO mutants, we used IHC to quantify the density of SST cIN efferent synapses, defined as the apposition of VGAT+ (vesicle GABA transporter) and gephyrin + puncta from an SST cIN axon within L1 of the S1 cortex at P8 (*Figure 5A*, black asterisks mark example puncta). We found that both *Sst-Nova1*-cKO (0.281±0.041 puncta/µm$^2$ *Sst-Nova1*-cKO vs 0.454±0.037 puncta/um$^2$ ctl, pVal = 0.003) and *Sst-Nova2*-cKO (0.197±0.016 puncta/um$^2$ *Sst-Nova2*-cKO vs 0.454±0.037 puncta/µm$^2$ ctl, pVal = <0.0001) exhibited a significant reduction in SST +synapses compared to control SST synapses within L1 (*Figure 5B*). To confirm the synaptic phenotype observed, we recorded the inhibitory outputs from SST cINs onto pyramidal cells in L2/3 and L5 using *Rosa26$^{LSL-hChR2}$* (Ai32) crossed with *Sst-Nova1*-cKO, *Sst-Nova2*-cKO, *Sst-Nova1/2*-dKO or SST-control mice (*Figure 5C*). While this experiment does not measure quantal postsynaptic currents, using a transgenic channelrhodopsin line ensures similar level of channel rhodopsin expression under mutant and control conditions. Using this strategy, we observed a significant reduction in the light evoked IPSC peak amplitude in *Sst-Nova1*-cKO (283±36 pA in *Sst-Nova1*-cKO vs 623±120 pA in ctl, pVal = 0.0037, *Figure 5D*), *Sst-Nova2*-cKO (340±85 pA in *Sst-Nova2*-cKO *vs* 766±211 pA ctl, pVal = 0.0021, *Figure 5E*), and *Sst-Nova1/2*-dKO (248.9 pA in *Sst-Nova1/2*-dKO vs 663 pA in ctl, *Figure 5F*) confirming that the anatomically observed reduction in synaptic output density is functionally significant in all mutants and did not differ between L2/3 and L5 pyramidal neurons, at least in the case of *Sst-Nova2*-cKO (*Figure 5—figure supplement 1J*).

We also investigated whether the density of excitatory synapses onto SST cINs is affected by the loss of *Nova1* or *Nova2*. We performed IHC for Vglut1 (vesicular glutamate transporter) and Homer1c on *Sst-Nova1*-cKO and *Sst-Nova2*-cKO dendrites within the S1 cortex at P8 (*Figure 5—figure supplement 1B*, black asterisks mark example puncta). We quantified the density of putative

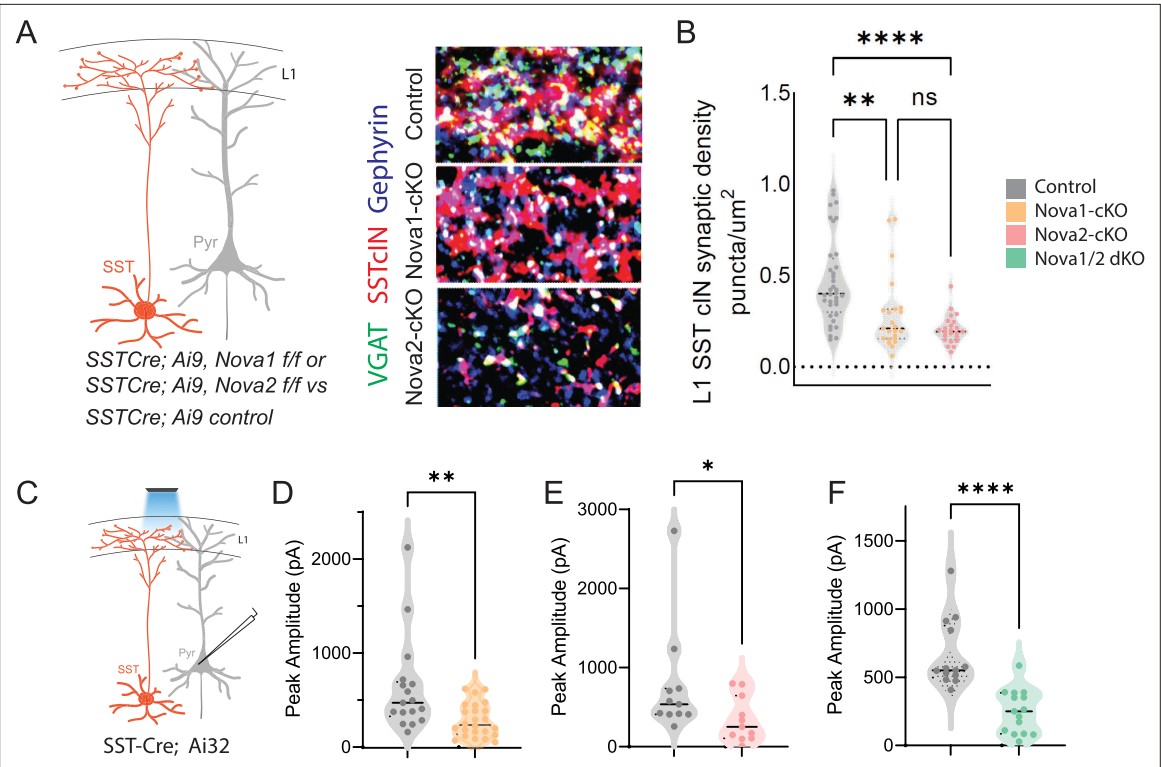

**Figure 5.** *SST-Nova1 and SST-Nova2* mutants have impaired afferent and efferent connectivity. (**A**) SST +cINs efferent structure: IHC of anti-RFP (red), anti-VGAT (green), and anti-Gephyrin (blue) to label the SST +cIN axonal synaptic puncta (RFP+/VGAT+/Gephyrin +puncta, white) in L1 S1 cortex of *Sst*-Ctl, *Sst-Nova1*-cKO, and *Sst-Nova2*-cKO mutant animals. (**B**) Quantification of the density of SST +cIN efferent synaptic puncta (RFP+/VGAT+/ Gephyrin+) in L1 S1 cortex of *Sst*-Ctl (*n=26, S1 cortex from 3 mice*), *Sst-Nova1*-cKO (*n=26, S1 cortex, from 3 mice*) and *Sst-Nova2*-cKO (*n=15, S1 cortex from 3 mice*) mutant animals. **pVal = 0.003, *Sst-Nova1*-cKO; ***pVal <0.0001, *Sst-Nova2*-cKO. (**C**) Schematic of channelrhodopsin (ChR2) experimental approach: *Sst*<sup>Cre</sup> control, *Sst-Nova1*-cKO, *Sst-Nova2*-cKO or *Sst-Nova1/2*-dKO mutant mice were crossed with the Ai32 reporter line that expresses ChR2 in a Cre-dependent manner. Blue light was delivered through the objective to record inhibitory response (IPSC) in neighboring excitatory neuron (grey). (**D–F**) Quantification of the peak IPSC amplitudes recorded in excitatory neurons following SST stimulation in *Sst-Nova1*-cKO (**D**), *Sst-Nova2*-cKO (**E**) and *Nova1/2*-dKO (**F**) (n=20 cells from 3 mice each; **pVal = 0.0037, *Sst-Nova1-cKO*; **pVal = 0.0021, *Sst-Nova2-cKO*, ***pVal <0.001).

The online version of this article includes the following figure supplement(s) for figure 5:

**Figure supplement 1.** Conditional loss of *Nova1/2* within SST +cINs impacts animal survival and disrupts their afferent synaptic connectivity.

excitatory synapses by the overlap of Vglut1 +and Homer1c+puncta onto mCherry + dendrites of SST cINs. We found that the number of putative excitatory afferent synapses onto *Sst-Nova1*-cKO and *Sst-Nova2-cKO* is significantly reduced compared to control SST cINs (0.144±0.016 puncta/μm² *Sst-Nova1*-cKO vs 0.207±0.022 puncta/μm² ctl, pVal = 0.028 and 0.137±0.013 puncta/μm² *Sst-Nova2*-cKO vs 0.207±0.022 puncta/μm² ctl, pVal = 0.012; *Figure 5—figure supplement 1C*). To examine whether these anatomical abnormalities observed in *Sst-Nova1*-cKO and *Sst-Nova2*-cKO mutants affected synaptic function, we performed whole-cell patch clamp recordings to measure miniature excitatory postsynaptic currents (mEPSCs) within SST cINs (*Figure 5—figure supplement 1D–I*). In accordance with the puncta analysis, both *Sst-Nova1*-cKO and *Sst-Nova2*-cKO exhibited significant reductions in the mEPSC frequency (*Sst-Nova1*-cKO: 1.16±0.08 Hz vs *Sst-Nova2*-cKO: 0.39±0.05 Hz vs ctl: 2.43±0.2 Hz, pVal = 0.0025, *Figure 5—figure supplement 1F*). In addition, we observed a significantly increased mEPSC amplitude in *Sst-Nova2*-cKO (*Sst-Nova2*-cKO: –40±15.7 pA vs *Sst-Nova1*-cKO: –30.12±13.15 pA vs ctl: –30.36±13.34 pA, pVal = 0.005, *Figure 5—figure supplement 1F* right and *Figure 5—figure supplement 1H*). Thus, while *Sst-Nova2*-cKO cINs have a striking reduction in their excitatory inputs, the remaining excitatory synapses are functionally stronger than *Nova1* or control cINs. Moreover, the intrinsic properties of both KO alleles were differentially affected. Specifically, we observed that the rheobase was significantly lower for *Sst-Nova2*-cKO compared with either controls or *Sst-Nova1*-cKO (*Sst-Nova2*-cKO: 25±3 pA vs ctl:120±25 pA vs *Sst-Nova1*-cKO: 70±15 pA; pVal = 0.01, *Supplementary file 1*). As rheobase is a measurement of

the minimum current required to produce an action potential, *Sst-Nova2*-cKO cINs are potentially compensating for the loss of excitatory synapses by lowering the minimal current amplitude required for depolarization. Altogether these results solidify the role of both Nova1 and Nova2 in the synaptic development of SST cINs. Furthermore, consistent with the AS analysis, these results suggest that within SST cINs Nova2 has a larger impact on the changes in synaptic connectivity compared to Nova1.

## Nova RNA binding proteins control-activity-dependent AS in SST cINS during development

Given that activity increases the expression level and nuclear localization of both Nova proteins, we hypothesized that their loss would result in changes in activity-dependent AS. To this end, we repeated our investigation of how Nova-dependent AS isoforms are altered in mutant mice. This time we examined the changes specifically following ECS within SST cINs during synaptogenesis in vivo. Two to 3 hr following ECS, we isolated SST cINs from *Sst-Nova1/2*-dKO mice (*Figure 6A*). Following augmentation of neuronal activity, we found that the loss of both *Nova* genes results in the differential splicing of 346 transcripts (FDR <0.05, $|\Delta\psi|\geq0.1$). These are comprised by 166 SE events (60 excluded and 106 included exons), 72 RI events (21 excluded and 51 included introns), 70 MXE events (33 excluded and 37 included exons), 9 A5 events (2 excluded and 7 included exons), and 29 A3 (20 excluded and 9 included exons; *Figure 6B*). Many of these genes were categorized into synaptic gene ontology categories both in AS and GE data with a small degree of overlap (*Figure 6C–E*). As demonstrated previously, many synaptic genes exhibited higher AS change level compared to GE (*Figure 6F* and *Figure 6—figure supplement 1E*). For example, the synaptic gene Nrxn1 was shown to have 4-fold difference in the AS level compared to GE (*Figure 6F* inset). Independent fluorescent RT-PCR amplifications with primers flanking the alternatively spliced segments confirmed the observed AS changes. We were able to validate 70% of targets tested. For example, we validated the activity-dependent inclusion of exon 4 in *Nrxn1*. As predicted from RNAseq, SST cINs subjected to acute increases in activity from *Sst-Nova1/2*-dKO animals, compared to control SST cINs, exhibit a significant reduction in the expression of *Nrxn1* exon 4 (*Figure 6G–H*). Similarly, we validated the activity-dependent inclusion of exon 14 in *Syngap1,* a gene associated in multiple disorders including epilepsy and important for excitatory post-synaptic function (*Figure 6—figure supplement 1C–D*). Both activity-mediated gene expression and splicing changes are partially abolished by *Nova1/2*-dKO (*Figure 6—figure supplement 1F–G*). A list of exon coverage and inclusion levels for synaptic genes is presented in *Supplementary file 2a*.

We found the majority of genes which undergo activity-induced Nova-dependent differential splicing were significantly enriched for GO categories such as pre-synaptic vesicular function, synapse organization, synaptic transmission, and neuronal growth (*Figure 6C* and *Figure 6—figure supplement 1A*). Many of the genes within these categories are known to have important functions for axon organization and synaptogenesis such as, *Nrxn1, Nrxn3, Plxna2,* and *Epha5*. Interestingly, the activity-dependent Nova AS targets were strikingly enriched for excitatory post-synaptic specializations such as, *Shank1*, *Syngap1*, *Dlg3*, *Grin1*, and *Gria1*. Furthermore, these genes are predicted to function together in a direct PPI network representing specific pre-synaptic and post-synaptic biological processes (direct network pVal = 0.0009, 10,000 permutations, *Figure 6—figure supplement 1B*). For example, the loss of *Nova* leads to an altered activity-dependent splicing program of multiple genes important to NMDA receptor-mediated signaling (*Grin1*) connected with PSD organization (e.g. *Dlg3*, *Shank1*) and $Ca^{2+}$-dependent signaling (e.g. *Hras, Rapgef1*) (*Figure 6—figure supplement 1B*).

In sum, the activity-mediated Nova-dependent AS changes within SST cINs are central for fine-tuning of synaptic development. We previously found that another important RNABP, Rbfox1, influences axonal development and also shuttles from the cytoplasm to the nucleus upon increase in activity in SST cINs (*Lee et al., 2009*; *Wamsley et al., 2018*). However, upon comparing the activity-dependent splicing programs within SST cINs of Rbfox1 (69 activity-dependent events) to Nova1/2 (346 activity-dependent events), we found Nova proteins control a much larger number of activity-dependent splicing events. This supports our hypothesis that Nova proteins are key players in the control of activity-dependent alternative splicing (*Figure 6—figure supplement 2*).

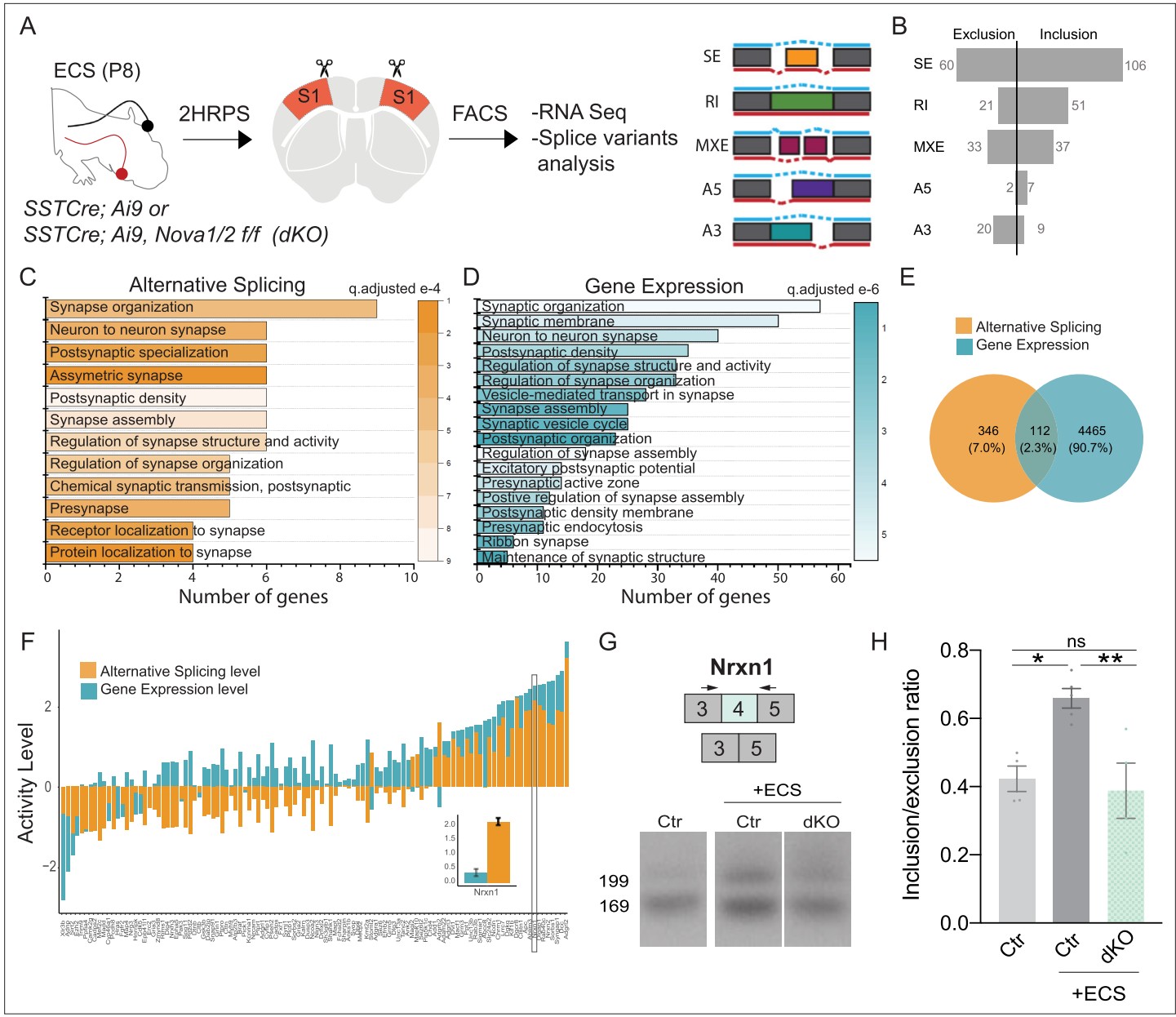

**Figure 6.** Nova RNA binding proteins control-activity-dependent AS in SST cINS during development. (**A**) Schematic of experimental approach: Control and *Sst-Nova1/2*-dKO P8 animals were subjected to ECS then the S1 cortex was isolated to FACS purify SST + cINs followed by RNAseq and splicing analysis. (**B**) Magnitude of activity-dependent splicing changes within *Sst-Nova1/2*-dKO subjected to ECS compared to Ctr *SST*- cINs subjected to ECS (FDR <0.5, fold <0.1 > ), depicting 166 differential spliced SE (106 SE included, 60 SE excluded), 72 differential spliced RI (51 RI included, 21 RI excluded), 70 differential spliced MXE (37 included, 33 excluded), 9 differential spliced A5 (7 included, 2 excluded), 29 differential spliced A3 (9 included, 20 excluded). (**C**) Synaptic gene ontology (GO) for the differentially spliced genes between ECS control vs ECS *Nova1/2*-dKO conditions. Color bar indicates adjusted q-value. (**D**) Synaptic gene ontology (GO) for the differentially expressed synaptic gene categories in the ECS control vs ECS *Nova1/2*-dKO conditions. (**E**) Number and percentage of overlap between all differentially expressed genes (FC >0.5, pVal <0.05) and alternatively splice genes. (**F**) Comparison of the activity level (Fold Change) of alternative splicing (AS) and gene expression (GE) amongst the shared genes that are both differentially expressed and differentially spliced. Inset shows that in the Nrxn1 gene AS level is larger (FC = 2.16) compared to the change in GE level (FC = 0.359). (**G**) Example RT-PCR validation of alternative splicing (AS) events of activity- and Nova1/2- dependent alternative exon usage within the gene *Nrxn1* (top), Gel image of RT-PCR product from the amplification of exon 3 to exon 5 within Sst-ctl cINs (Ctl) (left), ECS-treated Ctl (middle), and ECS- treated *Sst-Nova1/2*-dKO (right). (**H**) Quantification of RT-PCR AS events of *Nrxn1*. *pVal = 0.0194 Ctl vs Ctl + ECS; **pVal = 0.0087 Ctl + ECS vs *SST-Nova1/2*-dKO+ECS.

The online version of this article includes the following source data and figure supplement(s) for figure 6:

**Source data 1.** Gel showing Nrxn1 Exon 4 expression in different conditions.

*Figure 6 continued on next page*

*Figure 6 continued*

**Figure supplement 1.** Nova RNA binding proteins control-activity-dependent AS in SST cINs during development.

**Figure supplement 1—source data 1.** Gel showing Syngap Exon 14 expression.

**Figure supplement 2.** Nova1/2 controls the activity-dependent splicing of large and unique pool of mRNAs compared to Rbfox1 within SST cINs and SST-specific Nova2 AS genes overlap well with pan-cIN Nova2 AS genes.

## Augmenting activity in Nova2 KO fails to enhance SST inhibitory output

Activity increases both the expression of Nova proteins as well as synapse formation, while conversely loss of Nova function causes a striking decrease in synaptogenesis and SST inhibitory output. Moreover, from our analysis of SST cIN KOs, it was evident that of the two Nova proteins, *Nova2* has the more profound effect on the AS of genes involved in synaptogenesis. We therefore examined whether the loss of *Nova2* impaired the ability of augmented neuronal activity in SST cINs to promote the formation of efferent synaptic connectivity. To that end, we expressed NachBac in SST neurons in *Sst*$^{Cre}$::*Rose26*$^{LSL-hChR2}$ (Ai32) mice with *Nova2* deletions, compared with controls (**Figure 7A**). As previously shown, enhancing activity using NachBac resulted in increased Nova1/2 expression and localization into the nucleus in control mice (No Nova2-deletion, **Figure 7A** right). When we recorded from the pyramidal neurons in all conditions (control-No NachBac, control +NachBac, or *Nova2*-cKO+NachBac), we observed that enhancing activity in the *Nova2*-cKO did not result in an increase in inhibitory output of SST cINs (**Figure 7B**, right). This suggests that the activity-dependent changes of synaptic strength depend upon the presence of Nova2.

Conversely, we examined whether over-expression (OE) of Nova2 alone could phenocopy the observed changes in connectivity within SST cINs and whether that was affected by reducing the activity level of the cell (using Kir2.1). To that end, we either overexpressed *Nova2* alone specifically in SST + neurons using an AAV virus (AAV-Syn-Nova2-P2A-mCherry) in *Sst*$^{Cre}$ mice within the S1 cortex or in conjunction with Kir2.1 OE (**Figure 7C**). As in the case of increasing activity (either constitutively, NaChBac, or acutely, ECS), the nuclear localization of Nova was robustly increased when Nova2 was overexpressed (Nova2-OE) in the SST cINs (**Figure 7D–F**). The increased nuclear localization of Nova that was observed with the Nova2-OE was abolished when the activity of the cells was cell-autonomously reduced using KIR2.1.

We next also examined whether suppressing activity while overexpressing Nova2 impacts the inhibitory output of SST neurons (**Figure 7G** left). The dual expression of Nova2-OE and KIR2.1 within SST cINs prevented the small increase of peak IPSC amplitude observed with Nova2-OE alone. Perhaps most strikingly, as with our initial KIR2.1 experiment, the levels of Nova2 protein despite being constitutively OE were reduced in cells co-expressing KIR2.1 (**Figure 7E**). This provides strong evidence that the stability and nuclear localization of Nova protein is dependent on the level of basal activity within SST cINs. Therefore, a certain level of activity is needed to maintain Nova protein function, and conversely, Nova proteins are needed to mediate activity-dependent changes in alternative splicing of synaptic proteins.

## Discussion

In the present study, we have examined the interacting contributions of neuronal activity and the Nova RNABPs on synaptogenesis of SST cINs. Our analysis began with the observation that activity levels strongly influence the maturation of SST cINs. Acutely evoking activity during circuit integration with ECS resulted in both transcriptional and translational upregulation of Nova proteins and promoted their localization to the nucleus. This was accompanied by a striking change in both the GE and AS of synaptic genes and culminated in enhanced synaptogenesis within SST cINs. We then systematically examined the interdependence between these three observations.

Our results indicate that during circuit formation, activity levels within SST cINs correlate with changes in AS and together act to regulate the formation of afferent/efferent connectivity. These events appear to be tightly linked to Nova function, as the expression, localization and splicing activity of both Nova1 and Nova2 proteins are strongly modulated by activity. Examination of how splicing events are impacted by *Nova* single and compound KOs in SST cINs demonstrates that developmental

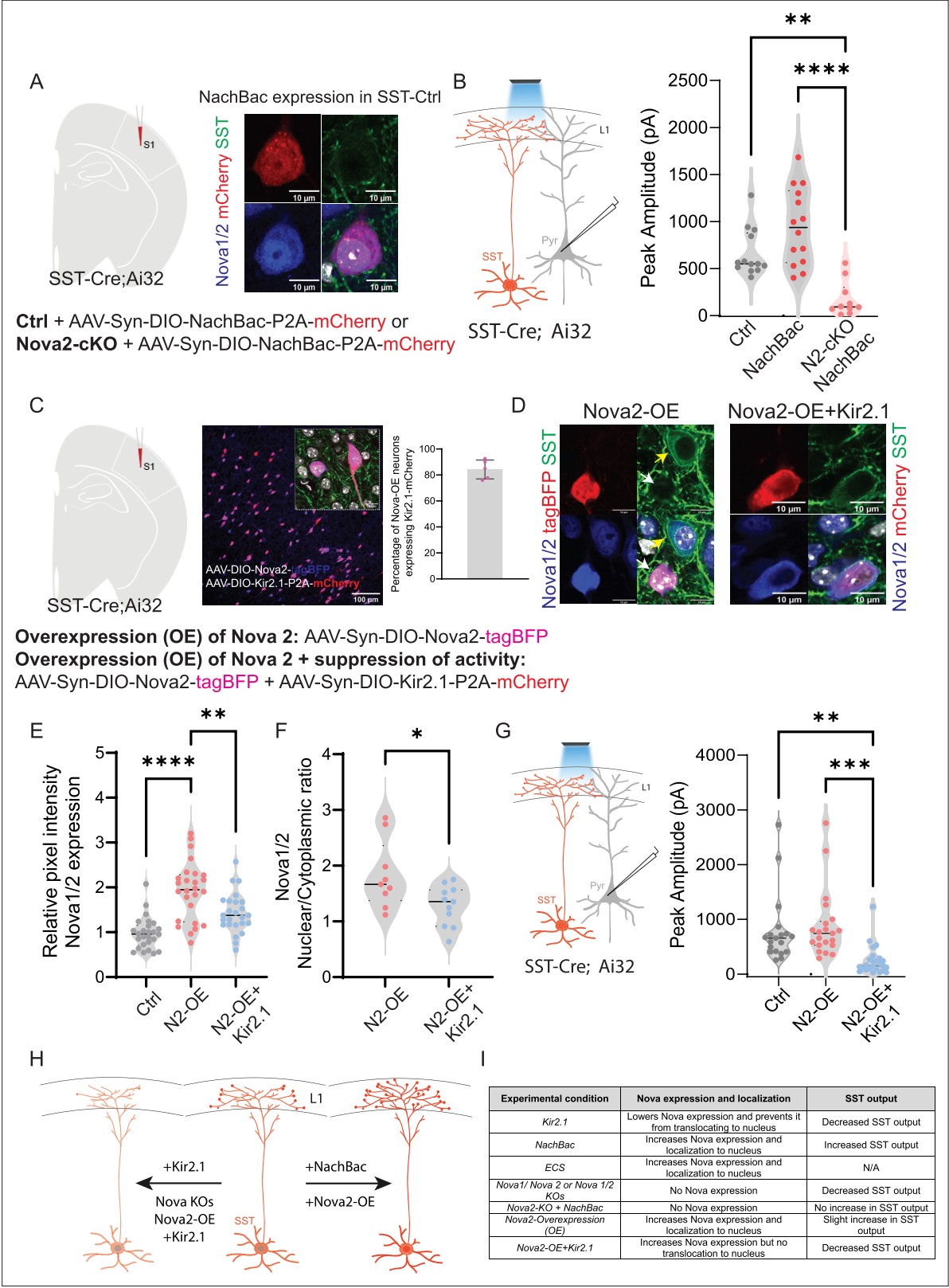

**Figure 7.** Augmenting activity in Nova2 KO fails to enhance SST inhibitory output. (**A**) Left, experimental model: Injection of AAV-Syn-DIO-NachBac-P2A-mCherry (activating) in either control mice or *Sst-Nova2*-cKO mice at P0 (analysis at P21). Right, example images showing the impact of NachBac activation (red) on Nova1/2 expression in controls. Note the translocation of Nova proteins (blue) to the nucleus (grey) in SST-cINs (green). Scale bar = 10 μm. (**B**) Left, schematic of the recording scheme: *SST$^{Cre}$*::Ai32 optogenetic activation and recording from L5 pyramidal neurons. Right, quantification

*Figure 7 continued on next page*

**Figure 7 continued**

of the peak IPSC amplitude recorded from pyramidal neurons under no NachBac control conditions (grey), NachBac injections in SST-Ctrl animals (red dots) or NachBac injections in *Sst-Nova2*-cKO animals (pink). (n=10–15 cells from each condition, N=3 mice; **≤0.01, ****≤0.001). (**C**) Left, experimental model: Overexpression (OE) of Nova2 using the AAV-Syn-DIO-Nova2-tagBFP virus was injected into $SST^{Cre}$::Ai32 mice either alone or while suppressing activity using Kir2.1 AAV-Syn-DIO-Kir2.1-P2A-mCherry. Middle, an image showing the co-expression of Nova2-tagBFP (blue) and Kir2.1-mCherry (red). Inset shows co-localization of both proteins in SST neurons. Scale bar of inset = 10 μm. Right, percentage of overlap between the two viruses in SST neurons, quantified as percentage of Nova2-OE neurons that also express Kir2.1-mCherry (~85%). (**D**) Left panels, Representative images of IHC against tagBFP (red), and Nova1/2 (anti-Nova1/2, blue) in SST-Nova2-OE cells in $SST^{Cre}$::Ai32 mice (green labels Ai32 expression). Right panels, SST-Nova2OE +KIR2.1 cell. Bottom right panels represent merged images. Note the exclusion of Nova proteins from the nucleus in Nova2-OE+Kir2.1 conditions. (**E**) Quantification of the relative pixel intensity of Nova1/2 expression in SST cINs (n=25/26 cells for each condition, pVal=**≤0.01, ****≤0.001). (**F**) Quantification of the Ratio of Nova1/2 localization within the nucleus to cytoplasm from Nova2OE SSt cINs (pink) and Nova2-OE+KIR2.1 (blue). (n=10 cells from 3 mice; pVal=*≤0.05). (**G**) Right, recording schematic. Left, Peak IPSC amplitude recorded from pyramidal neurons in response to optogenetic stimulation of SST-cINs in either the Nova2-OE condition or Nova2-OE+Kir2.1 condition (n=19–25 cells in each condition, pVal=**≤0.01, ***≤0.005). (**H**) Model of experimental findings: center is a cartoon wild type SST cIN depicting normal expression of Nova1/2 with the soma (red) whereas, on the left, the conditional loss of *Nova1, Nova2,* or the expression of KIR2.1 alone or dual overexpression of Nova2 and KIR2.1 results in the reduction in Nova expression and restricts Nova localization to the cytoplasm (In the case of KO animals the protein is lost completely). This effect is accompanied by a reduction in the connectivity of SST cINs. To the contrary, Expression of NaChBac and/or overexpression of Nova2 alone results in expression of Nova throughout the cell and nucleus and is accompanied by an increase in the SST cINs output. (**I**) Summary table of experimental findings in all conditions tested.

The online version of this article includes the following source data for figure 7:

**Source data 1.** Summary table for major experimental findings.

RNA splicing events in these cells are particularly impacted by the loss of *Nova2*. This is mirrored by the magnitude in reduction of excitatory input and inhibitory output within *Nova2* null SST cINs, as evidenced by a structural and functional decrease in their synaptic contacts. The relationship between activity and Nova2 function during development is interdependent. During these periods, boosting activity cell autonomously within *Nova2*-cKOs fails to increase the structural or physiological output of SST-cINs. Conversely, over-expression of Nova2 in SST cINs can enhance these activities but this phenomenon can be suppressed by simultaneous dampening their excitability. Together these findings demonstrate that activity is coupled to synaptogenesis in SST cINs by a mechanism involving Nova proteins. Whether these effects are regulated through their contributions to AS, GE or a combination of both remains to be determined.

With regards to AS in particular, Nova function is a core regulator of alternative splicing in many cell types, including SST cINs. It however represents only one of a host of RNABPs within the CNS. Indeed, a recent study demonstrated that within the mature brain many classes of neurons, including SST cINs, can be classified both by their expression levels of RNABPs and their corresponding repertoire of alternatively spliced mRNAs (*Furlanis et al., 2019*). Comparison of this work to our present findings illustrate that both the expression of RNABPs and the patterns of AS are strongly regulated across development, a phenomenon that may reflect developmental changes in neuronal activity. Consistent with this RNA binding splice factors have previously been shown to promote alternative splicing of synaptic proteins in response to neuronal depolarization and Ca$^{2+}$ signaling (*Eom et al., 2013*; *Mauger et al., 2016*; *Quesnel-Vallières et al., 2016*; *Vuong et al., 2016*), For example, previous research demonstrated that the splicing of neurexins, a gene family known to function in synaptogenesis, are mediated through the actions of the SAM68 splicing factor (*Iijima et al., 2011*). Similarly, It has also been illustrated that neuronal activity reduces the expression of the SRRM4 RNA-binding protein, which resulted in altered RNA splicing and a corresponding decrease in excitatory synapses (*Quesnel-Vallières et al., 2016*). As such AS represents a largely unexplored but central genetic mechanism, capable of directing cell-type development and synaptic formation specifically.

Understanding both the repertoire of splice factors and the cell-specific patterns of splicing across development will undoubtedly provide further insight into how AS influences cIN development. One could imagine systematically examining the role of these differential splice mRNA variants through combinatorial knockdown or over-expression. However, this would face enormous technical challenges, even if restricted to only those that are Nova-dependent. As we show here many of these genes have been shown to function together (PPI networks). As such AS appears to coordinately target specific biological mechanisms. Given that the abundance of the specific splice forms of different genes within SST cINs is relative rather than absolute, it appears that AS has been coopted

by development as an effective mechanism to fine-tune particular biological phenomena. The flexibility of AS to regulate the composition and levels of genes allows cells to adjust their biological function in accordance with both their identity and state (e.g. developmental period, neuronal activity, etc.). As a result, the abundance of specific splice forms co-varies as a function of both transcription and AS. Taken together, this argues that conditional removal of RNABPs, such as Nova2, provides an effective approach for understanding the role of AS within discrete cell types. Additionally, Nova proteins have a yet unexplored role in regulating gene expression, most likely through their ability to regulate the stability of RNA molecules.

In sum, our results show a clear interdependence between activity, Nova function and synaptic formation/strength in SST cINs. The interaction between activity and Nova function is bidirectional. Activity regulates the RNA, protein levels and intracellular localization of Nova proteins within SST cINs, while Nova proteins are in turn required for the activity-dependent regulation of synaptic formation and function (see model *Figure 7H*). When SST cIN activity is increased with ECS or with NaChBac expression, *Nova* transcripts as well as protein are upregulated and shuttled to the nucleus. The mechanisms for activity-dependent changes in Nova expression and localization are unknown. It is possible that the *Nova* gene loci may contain binding sites for immediate-early-genes (e.g. *cFOS, Jak/Jun, EGF*) or specific activity-dependent transcription factors (e.g. NPAS4, Satb1). With regard to control of its localization, previous work has discovered a nuclear-localization signal (NLS) within the Nova protein domains. It is however unknown whether their activation is also mobilized by splicing or post-translational modifications. For instance, *Rbfox1* undergoes activity-dependent mRNA splicing that results in exposure of an NLS and localization to the nucleus (*Lee et al., 2009*; *Wamsley et al., 2018*). Furthermore, our results indicate that activity itself regulates Nova2 RNA and protein stability. In the presence of KIR2.1, the levels of Nova protein appear to be dramatically reduced, even when Nova2 is over-expressed. In this latter context, clearly Nova2 levels are not constrained by mRNA production. These results indicate that the stability of Nova protein is at least partly dependent on activity. Taken together, these findings indicate that there exist multiple mechanisms by which cell activity is coupled to Nova function and AS within SST cINs.

We and others have shown that activity regulates programmed cell death (*Priya et al., 2018*; *Denaxa et al., 2018*; *Wong et al., 2018*). However, we observed no indication that the loss of *Nova2* impacted SST cIN survival. In addition, we observed that NaChBac and KIR2.1 could modulate synaptogenesis in SST cINs both during and after the peak of cell death in this region (data not shown). Conversely, the number of phenotypic changes observed in conditional *Nova* loss of function mutants suggests that these genes have effects beyond synaptogenesis. Nova2 also targets genes involved in protein trafficking to the membrane, cell-cell signaling, and neurotransmitter/ion channel function, indicating it influences multiple aspects of SST cIN maturation. In addition, prior work from the Darnell lab has demonstrated a role for Nova2 in both migration and axonal pathfinding within the cortex, spinal cord, and brain stem (*Saito et al., 2016*; *Yano et al., 2010*). Taken together clearly much remains to be understood concerning the role Nova proteins play during development in specific brain regions, circuits, and cell types. Indeed, given the broad expression of Nova proteins and the strong phenotypes associated with both conditional and global *Nova* loss of function, studies of this RNABP will no doubt provide further insights into their contribution to normal and disease brain function.

### Contact for reagent and resource sharing

Please contact GF or LAI for reagents and resources generated in this study.

## Materials and methods

**Key resources table**

| Reagent type (species) or resource | Designation | Source or reference | Identifiers | Additional information |
|---|---|---|---|---|
| Strain, strain background (*Mus musculus*) | SST-Cre | Jackson Laboratories | 13044 | |

*Continued on next page*

*Continued*

| Reagent type (species) or resource | Designation | Source or reference | Identifiers | Additional information |
|---|---|---|---|---|
| Strain, strain background (*Mus musculus*) | RCE-GFP | Jackson Laboratories | 032037-JAX | |
| Strain, strain background (*Mus musculus*) | tgLhx6;eGFP | MMRC | 000246-MU | |
| Strain, strain background (*Mus musculus*) | Nova1LoxP/LoxP | https://elifesciences.org/articles/00178 | Gift from Darnell Lab | |
| Strain, strain background (*Mus musculus*) | Nova2 LoxP/LoxP | https://elifesciences.org/articles/00178 | Gift from Darnell Lab | |
| Strain, strain background (*Mus musculus*) | TRE-Bi-SypGFP-tdTomato | Jackson Laboratories | 12345 | |
| Strain, strain background (*Mus musculus*) | Rosa-tTA LoxP/LoxP | Jackson Laboratories | 8600 | |
| Strain, strain background (*Mus musculus*) | Ai9 LoxP/LoxP | Jackson Laboratories | 7909 | |
| Strain, strain background (*Mus musculus*) | Ai32 LoxP/LoxP | Jackson Laboratories | 24109 | |
| Antibody | Anti-GFP, Chicken Polyclonal IgY | Abcam | Ab13970 | |
| Antibody | Anti-RFP (5 F8), Rat monoclonal | ChromoTek | 5 f8-100 | |
| Antibody | Anti-mCherry, Goat polyclonal | Origene | AB0040-200 | |
| Antibody | Anti-Somatostatin (YC7), Rat monoclonal | EMD Millipore | MAB354 | |
| Antibody | Somatostatin 14, Rabbit | Peninsula Labs | T-4103.0050 | |
| Antibody | Homer 1 c, Rabbit polyclonal | Synaptic systems | 160 023 | |
| Antibody | Vglut 1, Guinea pig polyclonal | Sigma | ab5905 | |
| Antibody | Gephyrin, Mouse IgG monoclonal | Synaptic systems | 147 011 | |
| Antibody | VGAT, Rabbit polyclonal | Synaptic systems | 131 003 | |
| Antibody | Nova1/2, Human polyclonal | pan-Nova (anti-Nova paraneoplastic human serum) | Gift from Darnell Lab | |
| Antibody | tagBFP, Rabbit polyclonal | Evrogen | AB233 | |
| Antibody | Anti-cFOS (4), Rabbit polyclonal | Santa Cruz Biotechnology | SC-52 | |
| Viral Vector | AAV-Syn-DIO-NachBac-P2A-mCherry | NYUAD | This paper | |
| Viral Vector | AAV-Syn-Kir2.1-P2A-mCherry | NYUAD | This paper | |
| Viral Vector | AAV-Syn-DIO-Nova2-tagBFP | NYUAD | This paper | |
| Viral Vector | VTKS2 Backbone | NYUAD | Addgene_170853 | |
| Software, algorithm | BEDTools | Quinlan Lab | v2.17.0 | |

*Continued on next page*

*Continued*

| Reagent type (species) or resource | Designation | Source or reference | Identifiers | Additional information |
|---|---|---|---|---|
| Software, algorithm | Picard tools | Broad Institute | http://broadinstitute.github.io/picard/ | |
| Software, algorithm | DESeq2 | Bioconductor | R studio package | |
| Software, algorithm | rMATS | Xing Lab | v3.0.9 | |
| Software, algorithm | Rstudio | Rstudio.com | Version 1.1.456 | |
| Software, algorithm | Custom code | This paper | https://github.com/IbrahimLab-23/Nova-proteins-and-synaptic-integration-of-Sst-interneurons; *Laboratory of Neural Circuits, 2023* | |
| Software, algorithm | ImageJ 2.0.0 Java 1.8.0_66 | National Institute of Health | https://imagej.net/; RRID:SCR_003070 | |
| Software, algorithm | | | | |
| Software, algorithm | Clampfit 10.7 (pClamp) | Molecular Devices | https://www.moleculardevices.com/products/software/pclamp.html; RRID:SCR_011323 | |
| Software, algorithm | | | | |
| Software, algorithm | Prism 9.1.2 | Graphpad Software | https://www.graphpad.com/; RRID:SCR_002798 | |
| Software, algorithm | | | | |
| Software, algorithm | | | | |
| Software, algorithm | Zen Blue | Zeiss | https://www.zeiss.com/microscopy/en_us/products/microscope-software/zen.html; RRID:SCR_013672 | |
| Software, algorithm | | | | |

## Mouse maintenance and mouse strains

All experimental procedures were conducted in accordance with the National Institutes of Health guidelines and were approved by the Institutional Animal Care and Use Committee of the NYU School of Medicine and Harvard Medical School. Generation and genotyping of *Sst^Cre* (JAX Stock No. 013044, *Taniguchi et al., 2011*), RCE^eGFP^(JAX Stock No. 032037, *Sousa et al., 2009*), Lhx6 BAC transgenic (referred to as TgLhx6;eGFP) (MMRC Stock No. 000246-MU, *Gong et al., 2003*), *Nova1^LoxP/LoxP^* (*Yuan et al., 2018*), *Nova2^LoxP/Lox^*(*Saito et al., 2019*), TRE-Bi-SypGFP-TdTomato (JAX Stock No. 012345, *Li et al., 2010*), and Ai9 *Rosa26^LSL-tdTomato^* (JAX Stock No. 007909), Ai32 *Rosa26^LSL-ChR2^* (JAX Stock No. 024109), *Rosa26^LSL-tTa^* (JAX Stock No. 008600). All mouse strains were maintained on a mixed background (Swiss Webster and C57/Bl6). The day of birth is considered P0. Information about the mouse strains including genotyping protocols can be found at http://www.jax.org/ and elsewhere (see above references).

## Immunochemistry and imaging

Embryos, neonate, juvenile, and adult mice were perfused inter cardiac with ice cold 4% PFA after being anesthetized on ice (neonates) or using sodium pentobarbital anesthesia in adults. Brains that were processed for immunofluorescence on slides were post-fixed and cryopreserved in 30% sucrose. Sixteen µm coronal sections were obtained using Cryostat (Leica Biosystems) and collected on superfrost coated slides, then allowed to dry and stored at –20 °C until use. For immunofluorescence, cryosections were thawed and allowed to dry for 5–10 min and rinsed in 1 x PBS. They were incubated at room temperature in a blocking solution of PBST (PBS-0.1%Tx-100) and 10% normal donkey serum (NDS) for 1 hr, followed by incubation with primary antibodies in PBS-T and 1% NDS at 4 °C overnight or 2 days. Samples were then washed 4 times with PBS-T and incubated with fluorescence-conjugated

secondary Alexa antibodies (Life Technologies) in PBS-T with 1% NDS at room temperature for 1 hr. Slides were incubated for 5 min with DAPI, washed three times with PBS-T. Then slides were mounted with Fluoromount G (Southern Biotech) and imaged.

Brains that were processed for free-floating immunofluorescence were first post-fixed in 4% PFA overnight at 4 °C. Fifty-μm-thickness brain slices were taken on a Leica vibratome and stored in a cryoprotecting solution (40% PBS, 30% glycerol and 30% ethylene glycol) at –20 °C. For immunofluorescence, floating sections were blocked for 1 hr at RT in normal donkey or goat serum blocking buffer and incubated for 2–3 days at 4 °C with primary antibodies in blocking buffer. Sections were washed 4x30 min at RT in PBST, incubated overnight at 4 °C with secondary antibodies and DAPI in blocking buffer, washed 4x30 min at RT in PBST before being mounted on super-frost plus glass slides. Primary antibodies are listed in Key Resource Table.

## Nova1/2 localization

To quantify the Nova localization in SST cINs, mCherry+/SST cIN, KIR2.1+/SST cINs or NaChBac+/SST cIN (n=27 cells from 3 mice each); control/SST cIN or ECS+/SST cINs (n=27 cells from 3 mice each); Nova2OE/SST cIN or Nova2OE +KIR2.1/SST cINs (n=20 cells from 3 mice) were binned into two categories based on the cell compartment Nova1/2 protein was localized to: Cytoplasmic restricted or Nuclear-expressing (comprised of nuclear restricted or whole soma localization). The number of Nuclear-expressing cells was then divided by the number of cytoplasmic restricted cells to obtain a ratio for Nova localization from either mCherry+/SST cIN or KIR2.1+/SST cINs. This was collected from at least three tissue sections from at least three animals.

## Electroconvulsive Shock

Electroconvulsive stimulation (ECS) was administered to animals with pulses consisting of 1.0 s, 50 Hz, 75 mA stimulus of 0.7ms delivered using the Ugo Basile ECT unit Model 57800, as previously described (*Guo et al., 2011*; *Ma et al., 2009*). Control/sham animals were similarly handled using the exact same procedure but without the current administration.

## Confocal imaging and synaptic puncta analysis

Animals were perfused as described above. Post-fixation incubation prior to cryopreservation was skipped. Cryostat sections (16 μm) were subjected to IHC as described above. Images were taken within the S1 cortex of at least three different sections from at least three different animals per genotype with a Zeiss LSM 800 laser scanning confocal microscope. Scans were performed to obtain four optical Z-sections of 0.33 μm each (totaling ~1.2 μm max projection) with a 63 x/1.4 Oil DIC objective. The same scanning parameters (pinhole diameter, laser power/offset, speed/averaging) were used for all images. Maximum projections of four consecutive 0.33 μm stacks were analyzed with ImageJ (NIH) puncta analyzer plugin (*Ippolito and Eroglu, 2010*) to count the number of individual puncta consisting of pre-synaptic and post-synaptic markers that are close enough together to be considered a putative synaptic puncta. Synaptic puncta density per image was calculated by normalization to total puncta acquired for each individual channel accounted in each image for each condition. Puncta Analyzer plugin for ImageJ is written by Barry Wark and is available for download (https://github.com/carina-block/Puncta-analyzer/tree/v1.0; *Wark et al., 2023*). Nova protein intensity was performed as: Cryostat sections of 20 μm were immunostained with goat anti-mCherry and human anti-pan Nova (from Darnell Lab). Images were analyzed using Fiji/ImageJ and Nova1/2 protein intensity levels were assessed normalized against area of the cells expressing the AAV.

## Electrophysiological recordings

### Slice preparation

Acute brain slices (300 μm thick) were prepared from P18-P22 mice. Mice were deeply anesthetized with isofluorane. The brain was removed and placed in ice-cold modified artificial cerebrospinal fluid (ACSF) of the following composition (in mM): 87 NaCl, 26 $NaHCO_3$, 2.5 KCl, 1.25 NaH2PO4, 0.5 CaCl, 4 MgCl2, 10 glucose, 75 sucrose saturated with 95% $O_2$, 5% $CO_2$ at pH = 7.4. Coronal sections were cut using a vibratome (Leica, VT 1200 S). Slices were then incubated at 34 C for 30 minutes and then stored at room temperature until use.

## Recordings

Slices were transferred to the recording chamber of an up-right microscope (Zeiss Axioskop) equipped with IR DIC. Cells were visualized using a 40 X IR water immersion objective. Slices were perfused with ACSF of the following composition (in mM): 125 NaCl, 25 NaHCO3, 2.5 KCl, 1.25 NaH$_2$PO$_4$, 2 CaCl$_2$, 1 MgCl$_2$, 20 glucose, saturated with 95% O$_2$, 5% CO$_2$ at pH = 7.4 and maintained at a constant temperature (31 °C) using a heating chamber. Whole-cell recordings were made from randomly selected tdTomato-positive SST interneurons or tdTomato negative pyramidal cells from layer II-III or layer V of the somatosensory cortex. Miniature synaptic currents were recorded in the presence of 1 uM TTX in ACSF. Recording pipettes were pulled from borosilicate glass capillaries (Harvard Apparatus) and had a resistance of 3–5 MΩ when filled with the appropriate internal solution, as reported below. Recordings were performed using a Multiclamp 700B amplifier (Molecular Devices). The current clamp signals were filtered at 10 KHz and digitized at 40 kHz using a Digidata 1550 A and the Clampex 10 program suite (Molecular Devices). Miniature synaptic currents were filtered at 3 kHz and recorded with a sampling rate of 10 kHz. Voltage-clamp recordings were performed at a holding potential of 0 mV. Current-clamp recordings were performed at a holding potential of –70 mV. Cells were only accepted for analysis if the initial series resistance was less than 40 MΩ and did not change by more than 20% throughout the recording period. The series resistance was compensated online by at least ~60% in voltage-clamp mode. No correction was made for the junction potential between the pipette and the ACSF.

Passive and active membrane properties were recorded in current clamp mode by applying a series of hyperpolarizing and depolarizing current steps and the analysis was done in Clampfit (Molecular Devices). The cell input resistance was calculated from the peak of the voltage response to a 50 pA hyperpolarizing 1 s long current step according to Ohm's law. Analysis of the action potential properties was done on the first spike observed during a series of depolarizing steps. Threshold was defined as the voltage at the point when the slope first exceeds a value of 20 V.s-1. Rheobase was defined as the amplitude of the first depolarizing current step at which firing was observed. Analysis of miniature inhibitory events was done using Clampfit's template search.

## Pipette solutions

Solution for voltage-clamp recordings from pyramidal cells (in mM): 125 Cs-gluconate, 2 CsCl, 10 HEPES, 1 EGTA, 4 MgATP, 0.3 Na-GTP, 8 Phosphocreatine-Tris, 1 QX-314-Cl and 0.4% biocytin, equilibrated with CsOH at pH = 7.3. Solution for current clamp recordings from SST cINs (in mM): 130 K-Gluconate, 10 KCl, 10 HEPES, 0.2 EGTA, 4 MgATP, 0.3 NaGTP, 5 Phosphocreatine and 0.4% biocytin, equilibrated with KOH CO2 to a pH = 7.3.

## *Nova2* OE/ *Nova2* OE +KIR2.1 experiment

*Sst*[Cre] mice crossed with Ai32 mice were injected at P0/1 with either AAV2/1-Syn-DIO-Nova2-tagBFP or together with AAV2/1-Syn-DIO-Kir2.1-mCherry at 1:1 ratio in the S1 cortex. Mice were perfused at P21, brains harvested, sucrose protected and sectioned on a freezing microtome (Leica) at 20 μm thickness as described above. Primary antibodies are listed in Key Resource Table.

## Optogenetic stimulation

Blue-light (470 nm) was transmitted to the slice from an LED placed under the condenser of an up-right microscope (Olympus BX50). IPSCs were elicited by applying single 1ms blue-light pulses of varying intensities (max. stimulation intensity ~0.33 mW/mm$^2$) and directed to L2/3 or L5 of the slice in the recording chamber. Light pulses were delivered every 5 s. The LED output was driven by a TTL output from the Clampex software of the pCLAMP 9.0 program suite (Molecular Devices).

## Isolation of cortical interneurons from the developing mouse cerebral cortex

Cortical interneurons were dissociated from postnatal mouse cortices (P8) as described (*Wamsley et al., 2018*). We collected at least 3–5 KO and 3–5 ctl brains and maintained overall balanced numbers of females and males within each condition, in order to avoid sex- related gene expression biases. Following dissociation, cortical neurons in suspension were filtered and GFP +or TdTomato + fate-mapped interneurons were sorted by fluorescence activated-cell sorting (FACS) on either a Beckman Coulter MoFlo

(Cytomation), BD FACSAria II SORP or Sony SY3200. Sorted cINs were collected and lyzed in 200 µl TRIzol LS Reagent, then thoroughly mixed and stored at –80 °c until further total RNA extraction.

## Nucleic acid extraction, RNA amplification, cDNA library preparation, and RNA sequencing

Total RNAs from sorted SST cINs (P8 mouse S1 cortices for *Figure 2*, *Figure 2—figure supplement 2C*, *Figure 4—figure supplement 1*, *Figure 4—figure supplement 2* and *Figure 5*) were extracted using TRIzol LS Reagent and PicoPure columns (if <20 K cells were recovered) or PureLink RNA Mini Kit (if >20 K cells were recovered), with PureLink DNase for on-column treatment, following the manufacturers' guidelines. RNA quality and quantity were measured with a Picochip using an Agilent Bioanalyzer and only samples with high quality total RNA were used (RIN: 7–10). 20 ng of total RNA was used for cDNA synthesis and amplification, using NuGEN Ovation RNA-Seq System V2 kit (NuGEN part # 7102). A total of 100 ng of amplified cDNA were used to make a library using the Ovation Ultralow Library System (NuGEN part # 0330). The samples were mulitplexed and subjected to 50-nucleotide paired-end read rapid with the Illumina HiSeq 2500 sequencer (v4 chemistry), to generate >50 million reads per sample. Library preparation, quantification, pooling, clustering and sequencing was carried out at the NYULMC Genome Technology Center. qRT-PCR (quantitative RT-PCR) was performed using SYBR select master mix (Thermo Fisher Scientific) on cDNA synthesized using SuperScript II reverse transcriptase and oligo(dT) primers.

List of RT- and qRT-PCR primers:

| Primer name | Sequence |
| --- | --- |
| Adam22-FAM-fw | CGTCGCCGTCCAGCTCGACCAGGGAATAATTGCCGGCACCAT |
| Adam22-Rv | GCGAGGTCTCCCATTTTCAC |
| Anks1b-FAM-Fw | CGTCGCCGTCCAGCTCGACCAGGCTCCCTAGACGTTCCTCAC |
| Anks1b-FAM-Fw | GGATGATGCTGCCAGTACTG |
| Sez6-FAM-Fw | CGTCGCCGTCCAGCTCGACCAGCCACCATCCACTTCTCCTGT |
| Sez6-Rev | GCTCCCTAGACGTTCCTCAC |
| Dlg3-FAM-Fw | CGTCGCCGTCCAGCTCGACCAGTTCCCTGGGTTAAGTGACGA |
| Dlg3-Rev | TCATCGTTGACTCGGTCCTT |
| Syngap1-FAM-Fw | CGTCGCCGTCCAGCTCGACCAGAACATCCAAAGGCAGCCAAG |
| Syngap1-Rev | GCCGGCTCACATAGAAAAGG |
| Prkrir-FAM-Fw | CGTCGCCGTCCAGCTCGACCAGGGGTTGAGAATTGTAGGAGAGC |
| Prkrir--Rev | CTGCTATGCGGGTTGTTCAA |
| Sorbs2-FAM-Fw | CGTCGCCGTCCAGCTCGACCAGCGATCGGAGCCAAGGAGTAT |
| Sorbs2-Rev | AGGCTTCTGTCTATGGAGGAC |
| Nrxn1-FAM-Fw | CGTCGCCGTCCAGCTCGACCAGACACCTGATGATGGGCGAC |
| Nrxn1-Rev | TGAAGCATCAGTCCGTTCCT |
| Ezh2-FAM-Fw | CGTCGCCGTCCAGCTCGACCAGTGAGAAGGGACCGGTTTGTT |
| Ezh2-Rev | GCATTCAGGGTCTTTAACGGG |
| Triobp-FAM-Fw | CGTCGCCGTCCAGCTCGACCAGACCCTAGCCAATGGACACAG |
| Triobp-Rev | CTTGAAGTTGAGCAGATCGGG |
| Itch-FAM-Fw | CGTCGCCGTCCAGCTCGACCAGTGCATTTCACAGTGGCCTTC |
| Itch-Rev | CCCATGGAATCAAGCTGTGG |

## Bioinformatics

Downstream computational analysis were performed at the NYULMC Genome Technology Center and at KAUST. All the reads were mapped to the mouse reference genome (mm10) using the STAR aligner (*Dobin et al., 2013*). Quality control of RNAseq libraries (i.e. the mean read insert sizes and their standard deviations) was calculated using Picard tools (v.1.126, RRID:SCR_006525) (http://broadinstitute.github.io/picard/). The Read Per Million (RPM) normalized BigWig files were generated using BEDTools (v2.17.0) (*Quinlan and Hall, 2010*) and bedGraphToBigWig tool (v4). For the SST cIN P8 ECS, approx. 60E6-80E6 reads were aligned per sample; for P8 *Sst-Nova1*-cKO, *Sst-Nova2*-cKO, *Sst-Nova1/2*-dKO, approx. 60E6-70E6 reads were aligned per sample; for P8 SST-cIN wt ECS and *Sst-Nova1/2*-dKO+ECS, approx. 60E6-80E6 reads were aligned per sample. The samples processed for downstream analysis were as follows: nine samples for SST cIN +ECS versus SST cIN ctl at P8 (4/5 samples per condition), sixsamples for *Sst-Nova1*-cKO removal versus SST cIN ctl (three samples per genotype), six samples for *Sst-Nova2*-cKO removal versus SST cIN ctl (three samples per genotype), six samples for *Sst-Nova1/2*-dKO removal versus SST cIN ctl (three samples per genotype), and seven samples for *Sst-Nova1/2*-dKO removal ECS versus SST cIN ctl ECS (fourcontrol samples, three KO samples). We performed differential expression analysis using DESeq2 R package for calculating the expression level of transcripts between different conditions. Genes with an adjusted p-value <0.05 and log fold change (FC)≥0.5 were considered differentially expressed.

We used rMATS (v3.0.9) to quantify the AS event types (i.e. Skipped exons (SE), alternative 3' splice sites (A3SS), alternative 5' splice sites (A5SS), mutually exclusive exons (MXE) and retained introns (RI)). rMATS uses a counts-based model, it detects AS events using splice junction and exon body counts and calculates an exon inclusion level value $\psi$ for each event in each condition. It then determines the differential $|\Delta\psi|$ value across conditions (cut-offs for significance were placed at FDR <0.05 and $|\Delta\psi|$≥0.1). To compare the level of similarity among the samples and their replicates, we used two methods: classical multidimensional scaling or principal-component analysis and Euclidean distance-based sample clustering. The downstream statistical analyses and generating plots were performed in Rstudio (Version 1.1.456) (http://www.r-project.org/).

To assess the enrichment for the Nova-binding motif in the differentially regulated exons we utilized rMAPS (*Park et al., 2016*). We utilized the raw output from rMATS analysis (6 RNAseq experiments of SST cINs +ECS vs SST cINs ctl) with significant splicing events cut off at FDR >50%. rMAPS performs position weight analysis to assess the enrichment of RNA-binding protein binding motifs in the exonic and flanking intronic regions of up-regulated or down-regulated exons and plots the motif density along with a given pValue in comparison to unregulated exons.

We performed GO analysis using the DAVID online Bioinformatics Resources 6.8 at FDR >0.05 (unless otherwise specified) (*Huang et al., 2009*) and tested PPI networks by utilizing DAPPLE at 10,000 permutations (*Rossin et al., 2011*). The GO categories were assigned to each group of genes, and after that, we used ClusterProfiler, the R function that helps with gene functional annotation and to perform GO enrichment analysis.

## Validation of SST-cINs AS activity-dependent exons by RT-PCR

Total RNAs from sorted cINs from wt/ctl SST cINs, ECS SST cINs, and ECS *Sst-Nova1/2*-dKO were extracted as described above and at least three independent biological replicates were used in each experiment. RT-PCR validation of regulated exons was performed as described before (*Han et al., 2014*). After denaturation, samples were run on 10% Novex TBE-Urea Gels (Thermo Fisher). Gels were directly scanned by ChemiDoc Imaging System (Bio-Rad) and quantified by ImageStudio program (Licor).

## Quantification and statistical analysis

No statistical method was used to pre-determine sample sizes, but our sample sizes were similar to those reported in previous publications in the field. In all figures: *, p-value <0.05; **, p-value <0.01; ***, p-value <0.001; ****, p-value <0.0001. Statistical analyses for motif enrichment were performed by rMAPS and differential alternative splicing changes were performed using rMATS. Percentages were compared with repeated t-tests in *GraphPad Prism* or *Rstudio*, and means ± (standard deviation, SD) are represented. Some statistical analyses and generating plots were performed in R environment (v3.1.1) (http://www.r-project.org/).

All values presented in the manuscript are average ± standard error of the mean (SEM). The statistical values for the intrinsic physiology are obtained using one-way ANOVA with Bonferroni correction for multiple comparisons between the different genotypes: Controls, *Nova1*-cKO, *Nova2*-cKO and *Nova1/2*-dKO (*p≤0.05, **p≤0.01, **p≤0.005). For the Channelrhodopsin output, we first determined if the data is normally distributed using Lilliefors test. In case of normal distribution, we performed student's t-test was used to compare Control vs *Nova1*-cKO, and Control vs *Nova2*-cKO (*p≤0.05, **p≤0.01, ***p≤0.005).

## Acknowledgements

We would like to thank the NYULMC Division of Advanced Research Technologies and their personnel: Mouse Genotyping Core (Jiali Deng and Jisen Dai); Cytometry and Cell Sorting Core (Kamilah Ryan, Keith Kobylarz, Yulia Chupalova, and Michael Gregory); Genome Technology Center (Adriana Heguy); and Applied Bioinformatics Laboratory (Aristotelis Tsirigos), which is supported in part by grant UL1 TR00038 from the National Center for Advancing Translational Sciences (NCATS), NIH. CCSC and GTC are supported by the Cancer Center Support Grant, P30CA016087, at the Laura and Isaac Perlmutter Cancer Center. We would also like to thank Yanjie Qiu and Marian Fernandez-Otero for helping with genotyping at Harvard Medical School. We would like to thank the extended Fishell Laboratory for critical reading of the manuscript. Work in the GF lab is supported by the following NIH grants: R01 NS081297, R01 MH071679, UG3 MH120096, P01 NS074972 and by the Simons Foundation SFARI.

## Additional information

### Competing interests

Jordane Dimidschstein, Gordon Fishell: Founder of Regal Therapeutics. The other authors declare that no competing interests exist.

### Funding

| Funder | Grant reference number | Author |
| --- | --- | --- |
| King Abdullah University of Science and Technology | BRF | Leena Ali Ibrahim |
| National Institutes of Health | R01 NS081297 | Gordon Fishell |
| National Institutes of Health | R01 MH071679 | Gordon Fishell |
| National Institutes of Health | UG3 MH120096 | Gordon Fishell |
| National Institutes of Health | P01 NS074972 | Gordon Fishell |
| Simons Foundation SFARI | | Gordon Fishell |

The funders had no role in study design, data collection and interpretation, or the decision to submit the work for publication.

### Author contributions

Leena Ali Ibrahim, Conceptualization, Data curation, Formal analysis, Supervision, Investigation, Visualization, Writing – original draft, Project administration, Writing – review and editing; Brie Wamsley, Conceptualization, Resources, Data curation, Formal analysis, Investigation, Visualization, Methodology, Writing – original draft, Project administration, Writing – review and editing; Norah Alghamdi, Formal analysis, Validation, Visualization, Writing – review and editing; Nusrath Yusuf, Data curation, Formal analysis, Validation, Writing – review and editing; Elaine Sevier, Data curation, Formal analysis, Writing – review and editing; Ariel Hairston, Alireza Khodadadi-Jamayran, Formal analysis; Mia Sherer, Xavier Hubert Jaglin, Data curation; Qing Xu, Lihua Guo, Yuan Yuan, Robert B Darnell, Resources; Emilia Favuzzi, Supervision; Jordane Dimidschstein, Methodology, Writing – original draft; Gordon

Fishell, Conceptualization, Supervision, Funding acquisition, Writing – original draft, Project administration, Writing – review and editing

### Author ORCIDs
Leena Ali Ibrahim  http://orcid.org/0000-0001-8255-3423
Qing Xu  http://orcid.org/0000-0001-9479-470X
Robert B Darnell  http://orcid.org/0000-0002-5134-8088
Gordon Fishell  http://orcid.org/0000-0002-9640-9278

### Ethics
All experimental procedures were conducted in accordance with the National Institutes of Health guidelines and were approved by the Institutional Animal Care and Use Committee of the NYU School of Medicine and Harvard Medical School (Protocol # IS00001269). All surgeries were performed under isoflurane anesthesia and every effort was made to minimize suffering.

### Decision letter and Author response
Decision letter https://doi.org/10.7554/eLife.86842.sa1
Author response https://doi.org/10.7554/eLife.86842.sa2

## Additional files

### Supplementary files
- MDAR checklist
- Supplementary file 1. Intrinsic properties of SST-Nova1 SST-Nova2 and SST-Nova12-dKO.
- Supplementary file 2. Exon coverage of synaptic genes.

### Data availability
Sequencing data has been deposited in GEO under accession code GSE143316. All processed RNA sequencing and splicing analysis data and sashimi plots can also be found at https://github.com/IbrahimLab-23/Nova-proteins-and-synaptic-integration-of-Sst-interneurons (copy archived at *Laboratory of Neural Circuits, 2023*).

The following dataset was generated:

| Author(s) | Year | Dataset title | Dataset URL | Database and Identifier |
|---|---|---|---|---|
| Wamsley B, Khodadadi-Jamayran A, Fishell G | 2022 | Nova proteins direct synaptic integration of somatostatin interneurons through activity-dependent alternative splicing | https://www.ncbi.nlm.nih.gov/geo/query/acc.cgi?acc=GSE143316 | NCBI Gene Expression Omnibus, GSE143316 |

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
