## [Editor Report]

This is an important study that explores the roles of a set of RNA binding proteins on the connectivity and development of a prominent class of neocortical interneurons. The authors provide convincing evidence that these proteins regulate alternative splicing of key effector genes in these neurons and regulate neuronal inputs and outputs in an activity-dependent manner.

---

## [Decision Letter]

**Decision letter after peer review:**

[Editors’ note: the authors submitted for reconsideration following the decision after peer review. What follows is the decision letter after the first round of review.]

Thank you for submitting your work entitled "Nova proteins direct synaptic integration of somatostatin interneurons through activity- dependent alternative splicing" for consideration by *eLife*. Your article has been reviewed by 3 peer reviewers, including Sacha B Nelson as the Reviewing Editor and Reviewer #1, and the evaluation has been overseen by a Senior Editor. The following individual involved in review of your submission has agreed to reveal their identity: Kevin Beier (Reviewer #2).

Our decision has been reached after consultation between the reviewers. Based on these discussions and the individual reviews below, we regret to inform you that your work will not be considered further for publication in *eLife*.

Summary

As you can see from the comments below, each of the reviewers found the topic important and thought that a number of the features of the study were well carried out and that there were some interesting results. Although each of the reviewers also raised concerns, there was some initial disagreement about how easily these could be addressed. After further discussion between the editors and reviewers, there was broad agreement that there are issues that make interpretation problematic and preclude publication without a significant amount of new work. Given the number of experiments that would be required to address the key concerns, and the *eLife* policy that revisions should entail only work that can be completed relatively quickly, we cannot recommend revision.

The core concerns (in part outlined below) are:

1) The necessity of Nova proteins for the effects of activity are not convincingly established;

2) Possible changes in gene expression levels are not adequately disentangled from changes in splicing;

3) The effect of activity on Nova protein stability/levels/activity are not adequately documented.

*Reviewer #1:*

This is a thorough and interesting study of the role of Nova1 and 2 RNA binding proteins (RBPs) in the activity-dependent development of Martinotti-type SST interneurons in the mouse cerebral cortex. The paper is likely to interest both those with a focused interest in neuronal splicing as well as those with more general interests in cortical circuit development. The paper will impact these fields both by revealing the importance of these specific RBPs as well as by providing a resource of activity-dependent splice variants which can be followed in these cells to understand either the mechanisms of activity-dependent control of these processes, or the downstream effects on synaptogenesis and circuit function. The major limitations of the present manuscript are that some findings are difficult to interpret, given the available data, and some of the presentation is hard to follow, perhaps reflecting incomplete editing of a prior revision.

1) The scientific point made in Figure 3B is unclear: If at P2, excitatory neurons have not yet turned on Nova1, does this really matter, does it really indicate any specificity? At P8 (more relevant to manipulation performed) is the difference between exc and inh. neurons significant? If significance is hard to assess, maybe this point (together with 3A, C should be left out).

2) p. 13: "Interestingly, SST-dKO mutants exhibited less altered splicing events than the single SST-Nova2mutant, suggesting that some inclusion and exclusion AS events are antagonistically directed by Nova1 and Nova2." This is indeed interesting, but it is difficult for the reader to assess the sensitivity and reproducibility of these measurements. Some estimate of the sensitivity and noise in the measurements that can then be used to predict the degree of overlap between the 3 KO conditions tested expected by chance is required if one is to conclude anything about these overlaps (including both the overlaps shown in Figure S4C-E and the reduced overlap in the number of events between Nova2 and the double KO). This is especially true since phenotypically, the Nova2 and double KO seem so similar (Figure 4).

3) The interpretation of Figure 6 depends upon knowing the relative level of overexpression of Nova2. Is the effect of KIR2.1 simply to reduce Nova2 levels back to control levels or is there a separate activity dependent regulation of Nova2 localization or activity which is at work here? Although ideally these experiments should have been designed in a way that permits this comparison, if they were not it would probably be sufficient to discuss this point and make clear to the reader that this is still unknown. The discussion does note: "In the presence of KIR2.1, the levels of Nova protein appear to be dramatically reduced, even when Nova2 is over-expressed" but it is not clear which data are being referred to. Figure 6 shows nuclear/cyto ratios but not overall protein levels.

*Reviewer #2:*

In the manuscript by Wamsley et al., the authors report that neuronal activity drives expression of Nova proteins in somatostatin interneurons in the cortex, and that Nova proteins direct both the development of inputs and outputs of these neurons through mediating alternative splicing of various transcripts. Though it is not explicitly shown that alternative splicing is the mechanism by which Nova proteins act, this is a reasonable hypothesis given the previously shown functions of Nova proteins. The investigators use a variety of methods to test their hypotheses and show how activity patterns can mediate the development of these cortical interneurons. The manuscript is both interesting and well-written. However, I have a few technical concerns about the work, and there are areas that I believe require further clarification.

The fact that mCherry is directly fused to Kir2.1 or NaChBac (or at least it appears so – information on these viral vectors is missing from the methods) may be problematic. Kir2.1 expression levels in neurons tend to be low, perhaps because the cell can only tolerate so much. I am not sure about NaChBac. This creates a problem when assessing axonal density in this manuscript e.g., Figures 1C and 1E. mCherry expression is dependent on both expression levels of the ion channel to which it is fused, as well as trafficking/localization. For the control experiment, cytoplasmic mCherry is used, which would have very different targeting/trafficking properties than membrane-localized protein. Therefore there are quite a few perhaps unintended variables in Figure 1 that the investigators should discuss.

I also thought that the data from these experiments should be normalized, for example to the numbers of infected cells, or expression levels per cell. Both metrics in Figure 1 (and all anatomical experiments throughout the manuscript) are dependent on how many cells are infected by the AAVs used, or how robustly the markers are expressed. Without this normalization, it is unclear how these results may be affected by technical artifacts.

It is confusing to me why synaptic physiology results from the double cKO animals are not presented in Figure 4. The authors note that these mice were physiologically abnormal and exhibit early lethality. Yet they were able to perform quality recordings from these animals (Figure S5). In that case I am curious what the puncta results would be from these double cKO animals (analogous to panels 4A-B and 4E-F).

For Figure 4, differences in IPSC amplitude could be driven by different numbers of infected cells, or more robust ChR2 expression in the control vs Nova1 or Nova2 cKO conditions. It would help to either normalize the expression of ChR2 (e.g., anatomical quantification). Also, the strength of individual synapses could be assessed by replacing extracellular ca^2+^ with Sr2+ to evoke quantal release from SST cINs.

The switching between a single cKO and double cKO animals was very confusing to me. If these mice were sufficiently problematic that the data should be presented in the supplemental and not main figures, I do not understand why Figure 5 was performed with the double cKO and not single cKO animals. This should be clarified.

The investigators make a note about AS events (relating to figure S4) that I feel is slightly misleading. They make the statement that "We found that the number of alterations in the SST-Nova2 AS events that overlap with SST-dKO is almost three times higher than that observed when comparing the overlap between SST-dKO and SST-Nova1." This is misleading as it was already noted that Nova2 loss results in about 3x more AS events than Nova1 loss. Thus it would be expected that, simply by this numeric difference, there would be 3x more overlap with the dKO condition. Indeed, for Nova1 25 overlap/124 total altered = 20%, and Nova2 62 overlap/339 total = 20%, or Nova1/Nova2 overlap 25/62 vs Nova1/Nova2 total 124/339 = 40%. Indeed this doesn't seem like a major point to emphasize, unless I am misunderstanding the intention of the authors.

Figure 5 seems highly focused though the results should have been broad given the nature of the assays being performed. The GO analysis in Figure S6 seems much more interesting than validating one gene via qPCR. Perhaps these data could be added to the main figure.

I was fairly satisfied with this manuscript until Figure 6, where I got confused.

Nova OE = more axons and synapses, Figure 6 (no information about inputs)

Nova cKO = fewer axons/synapses (Figure 4)

Kir2.1 = fewer axons/synapses (Figure 6)

Question: What would happen with activation (via ECS or NaChBac) in the Nova cKO mice?

Question: What is the effect of Nova cKO on basal cellular activity of cells normally expressing Nova?

It seems that Kir2.1, assuming the only effect is to hyperpolarize cells, is the dominant driver of function. However this is confusing to me. If the expression of Nova proteins is activity-dependent, it makes sense Kir2.1 would be epistatic to Nova protein expression. But it also appears to be the case that Kir2.1 functions downstream of Nova expression, as Kir2.1 can prevent the function of Nova OE. The only explanation I can think of is that reducing cellular activity (e.g., via Kir2.1) is the master regulator of this pathway – increased activity drives Nova protein expression, but reduced activity actively inhibits Nova expression and function. If this is the case, reducing activity thus is dominant over downstream pathways that typically signal axon/synapse growth (e.g., via Nova protein function).

However, how that would work in the context of the experiments performed is confusing.

NovaOE + Kir2.1 = blocks Nova function (presumably by reducing Nova protein levels), Figure 6. This makes sense if activity is necessary for transcription or translation of Nova proteins. But it is rather more confusing when considering that OE is though TRE-driven expression. Thus, one would have to argue either there are enhancer elements within the Nova protein that are activity-dependent, or rather, that Nova transcription is blocked without activity (though presumably Nova in the AAV context contains no introns, though this is not described in the methods). That, or somehow Nova protein is actively degraded on a rapid timescale in the absence of basal levels of activity. If this is the case, this should be explored, or explained. In general, the mode of activity as a master regulator, and Nova's potential role, should be more clearly explained or demonstrated.

*Reviewer #3:*

In their manuscript, Wamsley et al. present an interesting and novel set of claims which if true would be of appropriate impact for publication in *eLife*. They conclude that (1) during development, axonal and inhibitory synaptic density in SST-INs is mediated by activity, and (2) these axonal and synaptic changes are mediated in turn by expression and nuclear localization of splicing regulators Nova1 and Nova2 and (3) Nova1/2-mediated changes in mRNA splicing. While this is an appealing set of conclusions, some lines of evidence require further vetting and some inconsistencies required resolution before this should be considered for publication.

1. In Figure 1, the authors show phenotypic changes, axon and synapse reorganization, after activity manipulations using Kir2.1 and NaChBac. In order to truly show that Nova mediates Kir2.1 or NaChBac activity-induced changes in axon and synapse density – and to fully back the message claimed in the title – the authors should at least show that in the Nova KOs, NaChBac does not elicit the axon and synapse density increase shown in Figure 1. This question is somewhat investigated in Figure 6, but this experiment does not establish if the axon/synapse density decrease in KIR2.1 co-expressing animals is due to Nova down-regulation, or Nova down-regulation appears as one consequence of activity dampening, but has nothing to do with axon/synapse elimination. Alternatively, the authors might show in the Nova KOs that ECS-induced activity does not change axon synapse density. However, the current manuscript has not yet shown that ECS would eventually lead to phenotypic changes similar to those in Figure 1, nor if presumed splicing changes in the Kir2.1 and NaChBac experiments would be similar to those in ECS (see next point). Unless this can be thoroughly established, only the experiment involving Nova KOs and NaChBac-mediated activity would really establish the proposed pathway.

2. The manuscript focuses heavily on SST-INs and seems to imply a cell-autonomous effect, e.g. in Figures 1 and 3 the SST-Cre restricted manipulation of activity using Kir2.1 and NaChBac expression results in axonal and synaptic changes and Nova1/2 translocation. However, the use of broad ECS manipulations in other key figures leaves open the possibility that some observed changes are mediated by activity in excitatory or other IN types. Indeed, the authors acknowledge that Nova1 and 2 are both highly expressed in PV cINs as well as SST INs. Namely, in Figure 2, splicing changes in SST INs may be mediated by non-cell autonomous changes in activity distinct from the cell-autonomous effects in axonal and synaptic density or in Nova1/2 translocation. Before the authors can make any suggestion about SST-IN specific synaptic or axonal changes being mediated by changes in splicing, they at least should show SST-IN specific splicing changes using RNAseq in SST-Cre;AAV-flex-KIR2.1/NaChBac activity-manipulated animals, otherwise the SST-IN focus of the manuscript is somewhat unwarranted. The same applies for changes in expression of Nova1/2 (understandably here, potentially subtler cell-autonomous effects may not be easily detected with a WB) (Figure 3A-F). It would also be interesting to know the degree of overlap in up/down regulated genes between the experiments in Figure 2 and Figure 5, which was in contrast to Figure 2 performed with SST-Cre restricted KO of Nova1/2 (though, still with ECS).

3. Following onto this previous comment, overall throughout the manuscript a more detailed and quantitative treatment of the alternative splicing results would be informative and lend greater credibility to the results presented here. While the authors provide a GO analysis, a coarse summary of alternative splicing "event types", and anecdotally mention a few example genes by name, fuller documentation of the exact genes that have been alternatively spliced would allow better vetting of the results. Furthermore, annotation of Nrxn1 is not correct: exon numbering oddly seems to go against strand direction and the coordinates appear to point at exon 6 (or splice site 2) rather than 10. Previously this exon has been shown to be uniformly excised in SST cells, which is the opposite of what this study finds. This should be clarified. Finally, although the authors have data to show this, there is no mention of how gene expression level changes after each manipulation (ECS, KOs…). It would be greatly reassuring if the authors clearly present which genes experienced the largest splicing changes and whether their expression level has changed significantly. Currently, it is impossible to tell, for example, if we are looking at splicing changes in genes, which expression has majorly gone down. While splicing changes occurred, gene expression may reveal more immediate insights into the phenotypic outcomes, axon and synaptic reorganization. This would be especially important in the case of ECS experiments, but also interesting when Nova is overexpressed.

4. In Figure 4, authors should quantify SST-IN axonal density in Nova 1 and Nova 2 cKOs in addition to the synaptic changes, as changes in axonal density were also observed in the activity manipulations of Figure 1. It would be also interesting to know if the cell changes in excitatory input onto SST-INs observed in Figure 5 with SST-restricted Nova1 or 2 KO are observed with activity manipulation as well. Although a lack of Nova2 KO impact on cell survival is casually mentioned in Discussion, this should be quantified and documented for each KO condition.

5. Figure 2E is referenced in text, but not shown in figure.

6. Without further clarification, I find the presentation of Nova1/2 localization unconvincing. Figure 3G shows that Nova1/2 is either cytoplasmic or nuclear or both. However, for quantification, apparently only the nuclear/cytoplasmic ratio was used. It should be clearly stated if "both" cells were counted as "nuclear" or not, because outcomes can be very different. Furthermore, it should be clarified if any quantitative measure or cutoff was used to determine cytoplasmic versus nuclear localization. Were these blinded experiments?

[Editors’ note: further revisions were suggested prior to acceptance, as described below.]

Thank you for resubmitting your work entitled "Nova proteins direct synaptic integration of somatostatin interneurons through activity-dependent alternative splicing" for further consideration by *eLife*. Your revised article has been evaluated by Gary Westbrook (Senior Editor) and Sacha Nelson, Reviewing Editor.

The manuscript has been improved but there are some remaining issues that need to be addressed, as outlined below:

Please address each of the issues identified below. All of these changes should be able to be addressed with textual changes, and/or in one case a minor additional statistical analysis. The latter is concerned with Reviewer #2's point that results from 2/3 and 5 should be treated separately. In further consultation, the reviewers agreed that a simple test that the two are not different could be used to justify merging without further changes to figures or downstream analyses. It was also agreed that although a quantal measure from ChR2 experiments would be preferred because it affords better normalization, this issue could simply be acknowledged in the text.

*Reviewer #1 (Recommendations for the authors):*

The additional experiments go a large way toward addressing the key concerns raised by the reviewers. I have no further suggestions for improvement, other than two very minor clarifications.

102 RNABPs (i.e. PTBP1/2, FUS, ELAVL4, SRRM4, Rbfox1, FMR1, Nova1, Nova2) → 102 RNABPs (e.g. PTBP1/2, FUS, ELAVL4, SRRM4, Rbfox1, FMR1, Nova1, Nova2)

e.g. (as opposed to i.e.) makes it clear the listed RNABPs are examples.

"Sleepaway" should be replaced in the methods with a more generic description of the formulation or its components.

*Reviewer #2 (Recommendations for the authors):*

The authors performed a number of additional experiments to try to address the questions regarding the relationship between activity and Nova expression, which do improve the manuscript. It still is not crystal clear what the relationship is, but biology sometimes is not simple.

Several remaining issues:

The manuscript states that Nova proteins impact afferent and efferent connectivity through alternative splicing. I would say the manuscript falls a bit short of making that causal relationship. The data imply it, but no experiments were done specifically altering splice events individually or globally and assessing the impacts on connectivity.

Authors report measuring mPSCs, however, no TTX was included in recording preparations. Therefore these would be better described as spontaneous PSCs, not minis.

Why were results from L2/3 and L5 cells merged? They should also be assessed independently.

A quantal measure of elicited PSC amplitude would still be a superior measure to the non-normalized version presented, even if the number of cells expressing ChR2 should theoretically not change.

The authors make the statement "we found a small decrease in the ratio of nucleus to cytoplasmic Nova protein within SST cINS injected with Kir2.1". It is unclear why this decrease is noted as small: the ratio (4.75 to.265) is approximately 20:1, similar to but larger than that for NaChBac (4.75 to 40.91, 1:10) listed as substantial. It is not clear why this wording was used.

Differences in significance for GO results correlate with the number of altered AS genes identified. This perhaps is not a function of the biological significance of those genes identified in this way, but just the number of genes used as inputs.

For S6F and S6H, it is unclear to me how a pval = 0.0001 could be obtained from these plots, at least with the values given in the text vs. the figure. Also, ** is noted, vs. *** which might be expected. I'm just confused by this. Is this also a difference in SEM vs. SD?

---

## [Author Response]

[Editors’ note: the authors resubmitted a revised version of the paper for consideration. What follows is the authors’ response to the first round of review.]

The core concerns (in part outlined below) are:1) The necessity of Nova proteins for the effects of activity are not convincingly established;

In addition to better organizing how electroconvulsive shock regulates Nova levels and localization (figure 3), as well as a complete reorganization of the loss of function results (figure 4), we have added experiments in figure 5 on how the loss of Nova genes impacts synaptic function and the loss of Nova genes blunts the splicing changes normally induced by ECS (Figure 6). Finally, in entirely new data we explore in Figure 7 the epistatic interdependence of activity and Nova2 with regard to their ability to regulate synaptic formation and strength is explored.

2) Possible changes in gene expression levels are not adequately disentangled from changes in splicing;

Given the substantial changes seen in both Nova cKOs, truly disentangling the changes mediated by Nova in gene expression versus alternatively splicing is a massively daunting undertaking that goes beyond the reasonable scope of any single paper. That said we have done a good faith effort to catalog these changes in both Nova knockouts (Figure 5) and by comparing gene expression changes after ECS in wild type versus Nova cKOs. As the reviewers can appreciate, gene expression can by itself be affected by alterative splicing either by directly changing the stability of the RNA transcript or by Nova stabilization of RNAs themselves. While describing these changes is possible and has now been done in the revised manuscript, accounting for what aspects of Nova function is attributable to GE versus AS is task that will take years of experimentation to sort out. We acknowledge this in the revised discussion and point out specifically how one might undertake experiments aimed at fully addressing these important issues.

3) The effect of activity on Nova protein stability/levels/activity are not adequately documented.

These issues are now addressed in revised figures 4-7 as discussed above and below. In short, we have gone to an extraordinary length to explore these questions in single and double KOs. Moreover, we have examined the interdependence between Nova2 and activity through reciprocal epistatic experiments in Figure 7 (NaChBacmediated SST cIN cell autonomous increase in activity in Nova2 cKOs versus Nova2 OE in the context of cell autonomous Kir2.1 activity suppression). Specially, while the question of Nova2 protein levels dependence on activity is demonstrated to be independent of transcription, and the two-way relationship between activity and Nova2 is explored, we did not undertake the logistically complex experiments to examine Nova1 and Nova1/2 OE and LOF experiments in the context of activity modulation. Our decision not to undertake these further experiments was both due to the magnitude of the task, as well as because our LOF analysis indicates that Nova1 has a lesser function in SST cINs, which based on the double KO analysis is partly antagonistic to Nova2.

Reviewer #1:This is a thorough and interesting study of the role of Nova1 and 2 RNA binding proteins (RBPs) in the activity-dependent development of Martinotti-type SST interneurons in the mouse cerebral cortex. The paper is likely to interest both those with a focused interest in neuronal splicing as well as those with more general interests in cortical circuit development. The paper will impact these fields both by revealing the importance of these specific RBPs as well as by providing a resource of activity-dependent splice variants which can be followed in these cells to understand either the mechanisms of activity-dependent control of these processes, or the downstream effects on synaptogenesis and circuit function. The major limitations of the present manuscript are that some findings are difficult to interpret, given the available data, and some of the presentation is hard to follow, perhaps reflecting incomplete editing of a prior revision.

We hope that in the extensive revision we have done in streamlining and thoroughly editing our manuscript has clarified the data and our narrative.

1) The scientific point made in Figure 3B is unclear: If at P2, excitatory neurons have not yet turned on Nova1, does this really matter, does it really indicate any specificity? At P8 (more relevant to manipulation performed) is the difference between exc and inh. neurons significant? If significance is hard to assess, maybe this point (together with 3A, C should be left out).

We agree with the reviewer that the enrichment in inhibitory versus excitatory cells is of only secondary importance to this study and have now relegated this data to Supplemental Figure 3C. Although Nova proteins are only slightly enriched in cINs vs excitatory neurons during the relevant time period, this serves at least to provide an indication of their relative abundance.

2) p. 13: "Interestingly, SST-dKO mutants exhibited less altered splicing events than the single SST-Nova2mutant, suggesting that some inclusion and exclusion AS events are antagonistically directed by Nova1 and Nova2." This is indeed interesting, but it is difficult for the reader to assess the sensitivity and reproducibility of these measurements. Some estimate of the sensitivity and noise in the measurements that can then be used to predict the degree of overlap between the 3 KO conditions tested expected by chance is required if one is to conclude anything about these overlaps (including both the overlaps shown in Figure S4C-E and the reduced overlap in the number of events between Nova2 and the double KO). This is especially true since phenotypically, the Nova2 and double KO seem so similar (Figure 4).

In the analytical splicing framework of this paper, we selected rMATS specifically for its superior performance compared to other methods (i.e. DEXseq, MISO, Leafcutter) to be sensitive enough to detect robust splicing events shared among the replicates of mutants compared to controls. rMATs is also shown to reduce random sampling noise that might introduce false positive or skew the inclusion and exclusion events (even three years later, rMATs remains the tool of choice in the field). We think this is exemplified not only in the obvious biological overlaps between mutant splicing profiles mentioned here but also in our success in verifying rMATA splicing predictions.

3) The interpretation of Figure 6 depends upon knowing the relative level of overexpression of Nova2. Is the effect of KIR2.1 simply to reduce Nova2 levels back to control levels or is there a separate activity dependent regulation of Nova2 localization or activity which is at work here? Although ideally these experiments should have been designed in a way that permits this comparison, if they were not it would probably be sufficient to discuss this point and make clear to the reader that this is still unknown. The discussion does note: "In the presence of KIR2.1, the levels of Nova protein appear to be dramatically reduced, even when Nova2 is over-expressed" but it is not clear which data are being referred to. Figure 6 shows nuclear/cyto ratios but not overall protein levels.

Our impression both in terms of the protein level (based on immunocytochemistry), as well as synapse formation/function, is that the Kir2.1 reduces it below normal physiological levels. Whether this is through reduced translation or localization is impossible to know. However, the ability to suppress supernumerary synapse formation in NOVA2 OE through Kir2.1 expression indicates it could act entirely through its influence on translation and localization. We have now made these points in the revised text.

Reviewer #2:In the manuscript by Wamsley et al., the authors report that neuronal activity drives expression of Nova proteins in somatostatin interneurons in the cortex, and that Nova proteins direct both the development of inputs and outputs of these neurons through mediating alternative splicing of various transcripts. Though it is not explicitly shown that alternative splicing is the mechanism by which Nova proteins act, this is a reasonable hypothesis given the previously shown functions of Nova proteins. The investigators use a variety of methods to test their hypotheses and show how activity patterns can mediate the development of these cortical interneurons. The manuscript is both interesting and well-written. However, I have a few technical concerns about the work, and there are areas that I believe require further clarification.The fact that mCherry is directly fused to Kir2.1 or NaChBac (or at least it appears so – information on these viral vectors is missing from the methods) may be problematic. Kir2.1 expression levels in neurons tend to be low, perhaps because the cell can only tolerate so much. I am not sure about NaChBac. This creates a problem when assessing axonal density in this manuscript e.g., Figures 1C and 1E. mCherry expression is dependent on both expression levels of the ion channel to which it is fused, as well as trafficking/localization. For the control experiment, cytoplasmic mCherry is used, which would have very different targeting/trafficking properties than membrane-localized protein. Therefore there are quite a few perhaps unintended variables in Figure 1 that the investigators should discuss.

Kir2.1 and NachBac are separated by a 2A sequence, and we have now clarified this.

I also thought that the data from these experiments should be normalized, for example to the numbers of infected cells, or expression levels per cell. Both metrics in Figure 1 (and all anatomical experiments throughout the manuscript) are dependent on how many cells are infected by the AAVs used, or how robustly the markers are expressed. Without this normalization, it is unclear how these results may be affected by technical artifacts.

Now corrected. We analyzed about 30 cells per image per three images per replicate. When we did normalize to number of cells infected it did not change the significant results, which is now included in the manuscript, thus the density of infected cells did not change the cell–specific changes in Nova localization and expression.

It is confusing to me why synaptic physiology results from the double cKO animals are not presented in Figure 4. The authors note that these mice were physiologically abnormal and exhibit early lethality. Yet they were able to perform quality recordings from these animals (Figure S5). In that case I am curious what the puncta results would be from these double cKO animals (analogous to panels 4A-B and 4E-F).

This physiology data for the Nova1/2-dKO has now been included.

For Figure 4, differences in IPSC amplitude could be driven by different numbers of infected cells, or more robust ChR2 expression in the control vs Nova1 or Nova2 cKO conditions. It would help to either normalize the expression of ChR2 (e.g., anatomical quantification). Also, the strength of individual synapses could be assessed by replacing extracellular ca^2+^ with Sr2+ to evoke quantal release from SST cINs.

ChR2 was expressed using the Ai32 reporter mouse line and hence the expression was uniform in terms of number of cells in all conditions.

The switching between a single cKO and double cKO animals was very confusing to me. If these mice were sufficiently problematic that the data should be presented in the supplemental and not main figures, I do not understand why Figure 5 was performed with the double cKO and not single cKO animals. This should be clarified.

We were concerned whether Nova1 and Nova2 compensated for each other. It seems that Nova2 is the central Nova required in SST cINs and based on the double KO these two may function somewhat antagonistically. We think that while all of this is of interest to the reader, it thoroughly justifies using Nova2 OE and cKO as the focus of the paper and by proxy figure 7.

The investigators make a note about AS events (relating to figure S4) that I feel is slightly misleading. They make the statement that "We found that the number of alterations in the SST-Nova2 AS events that overlap with SST-dKO is almost three times higher than that observed when comparing the overlap between SST-dKO and SST-Nova1." This is misleading as it was already noted that Nova2 loss results in about 3x more AS events than Nova1 loss. Thus it would be expected that, simply by this numeric difference, there would be 3x more overlap with the dKO condition. Indeed, for Nova1 25 overlap/124 total altered = 20%, and Nova2 62 overlap/339 total = 20%, or Nova1/Nova2 overlap 25/62 vs Nova1/Nova2 total 124/339 = 40%. Indeed this doesn't seem like a major point to emphasize, unless I am misunderstanding the intention of the authors.

We agree with the reviewer’s point that the overlap could simply be due to increased number of splicing events and a statistic correlation would be informative in the overlap. However, It should also be noted that this finding is more of an independent biological replication and confirmation of what has been seen in similar bulk and excitatory neuronal analysis – however in this current manuscript version as noted above we have thus focused our analysis on the role of Nova2, with the discussion and data from Nova1 and the double KO being included for completeness.

Figure 5 seems highly focused though the results should have been broad given the nature of the assays being performed. The GO analysis in Figure S6 seems much more interesting than validating one gene via qPCR. Perhaps these data could be added to the main figure.

The central focus of this paper is on the joint role of activity and Nova genes in synaptogenesis in SST cINs. We have now included the GO analysis both for GE and AS in the main text and supplementary figures.

I was fairly satisfied with this manuscript until Figure 6, where I got confused.Nova OE = more axons and synapses, Figure 6 (no information about inputs)Nova cKO = fewer axons/synapses (Figure 4)Kir2.1 = fewer axons/synapses (Figure 6)Question: What would happen with activation (via ECS or NaChBac) in the Nova cKO mice?Question: What is the effect of Nova cKO on basal cellular activity of cells normally expressing Nova?

We have now included this experiment in Figure 7. We investigated whether activation using NachBac in the Nova2-KO will compensate for the lack of Nova2. Interestingly, we found that NachBac was unable to induce activity dependent changes in synaptic output without the presence of Nova2. On the other hand, reducing activity using Kir2.1 while overexpressing Nova2, brought the physiological output below baseline. This suggests that Nova is needed to mediate the activity dependent changes, as well as a basal level of activity is needed to translocate Nova to the nucleus which is needed for Nova to function.

It seems that Kir2.1, assuming the only effect is to hyperpolarize cells, is the dominant driver of function. However this is confusing to me. If the expression of Nova proteins is activity-dependent, it makes sense Kir2.1 would be epistatic to Nova protein expression. But it also appears to be the case that Kir2.1 functions downstream of Nova expression, as Kir2.1 can prevent the function of Nova OE. The only explanation I can think of is that reducing cellular activity (e.g., via Kir2.1) is the master regulator of this pathway – increased activity drives Nova protein expression, but reduced activity actively inhibits Nova expression and function. If this is the case, reducing activity thus is dominant over downstream pathways that typically signal axon/synapse growth (e.g., via Nova protein function).However, how that would work in the context of the experiments performed is confusing.

As now shown in Figure 7, as well as discussed extensively in the text, neuronal activity and Nova work in unison to control synaptogenesis. Activity controls the levels and localization of Nova2, while Nova2 is required for activity to promote synaptogenesis. As such they are not epistatic to one another in the tradition sense but rather act to coordinate synapse formation through AS and GE.

NovaOE + Kir2.1 = blocks Nova function (presumably by reducing Nova protein levels), Figure 6. This makes sense if activity is necessary for transcription or translation of Nova proteins. But it is rather more confusing when considering that OE is though TRE-driven expression. Thus, one would have to argue either there are enhancer elements within the Nova protein that are activity-dependent, or rather, that Nova transcription is blocked without activity (though presumably Nova in the AAV context contains no introns, though this is not described in the methods). That, or somehow Nova protein is actively degraded on a rapid timescale in the absence of basal levels of activity. If this is the case, this should be explored, or explained. In general, the mode of activity as a master regulator, and Nova's potential role, should be more clearly explained or demonstrated.

There is no question that the interaction between activity and Nova are multifaceted.

Activity regulates the levels of Nova transcript, protein and localization. Nova in turn regulates AS and GE and synaptogenesis and likely other cellular functions in both SST and other neuronal types. We feel figure 7 provides a parsimonious explanation for how activity and Nova2 are coordinated but clearly with regard to both activity and Nova they influence SST cIN function more broadly.

Our data suggest that activity is needed for the localization of Nova to the nucleus which is suppressed using Kir2.1 (even if we overexpress Nova2). In this revised manuscript we constructed a new AAV, where TRE elements are not involved. We have spent three years expanding on this story and clarifying it and hope the reviewers are satisfied with what we consider to be a good faith effort to focus and refine our findings.

Reviewer #3:In their manuscript, Wamsley et al. present an interesting and novel set of claims which if true would be of appropriate impact for publication in eLife. They conclude that (1) during development, axonal and inhibitory synaptic density in SST-INs is mediated by activity, and (2) these axonal and synaptic changes are mediated in turn by expression and nuclear localization of splicing regulators Nova1 and Nova2 and (3) Nova1/2-mediated changes in mRNA splicing. While this is an appealing set of conclusions, some lines of evidence require further vetting and some inconsistencies required resolution before this should be considered for publication.1. In Figure 1, the authors show phenotypic changes, axon and synapse reorganization, after activity manipulations using Kir2.1 and NaChBac. In order to truly show that Nova mediates Kir2.1 or NaChBac activity-induced changes in axon and synapse density – and to fully back the message claimed in the title – the authors should at least show that in the Nova KOs, NaChBac does not elicit the axon and synapse density increase shown in Figure 1. This question is somewhat investigated in Figure 6, but this experiment does not establish if the axon/synapse density decrease in KIR2.1 co-expressing animals is due to Nova down-regulation, or Nova down-regulation appears as one consequence of activity dampening, but has nothing to do with axon/synapse elimination. Alternatively, the authors might show in the Nova KOs that ECS-induced activity does not change axon synapse density. However, the current manuscript has not yet shown that ECS would eventually lead to phenotypic changes similar to those in Figure 1, nor if presumed splicing changes in the Kir2.1 and NaChBac experiments would be similar to those in ECS (see next point). Unless this can be thoroughly established, only the experiment involving Nova KOs and NaChBac-mediated activity would really establish the proposed pathway.

In our revised paper, Figures 6 and 7 are aimed to address the precise questions raised by the reviewer. In short, the changes in AS and GE after ECS in wild type versus Nova1/2-dKO are compared in figure 6. In figure 7, the mutual dependence between activity increase in Nova2 cKO, as well as activity decrease in Nova2 OE are compare with regard to their complementary effects of synaptic structure/function.

2. The manuscript focuses heavily on SST-INs and seems to imply a cell-autonomous effect, e.g. in Figures 1 and 3 the SST-Cre restricted manipulation of activity using Kir2.1 and NaChBac expression results in axonal and synaptic changes and Nova1/2 translocation. However, the use of broad ECS manipulations in other key figures leaves open the possibility that some observed changes are mediated by activity in excitatory or other IN types. Indeed, the authors acknowledge that Nova1 and 2 are both highly expressed in PV cINs as well as SST INs. Namely, in Figure 2, splicing changes in SST INs may be mediated by non-cell autonomous changes in activity distinct from the cell-autonomous effects in axonal and synaptic density or in Nova1/2 translocation. Before the authors can make any suggestion about SST-IN specific synaptic or axonal changes being mediated by changes in splicing, they at least should show SST-IN specific splicing changes using RNAseq in SST-Cre;AAV-flex-KIR2.1/NaChBac activity-manipulated animals, otherwise the SST-IN focus of the manuscript is somewhat unwarranted. The same applies for changes in expression of Nova1/2 (understandably here, potentially subtler cell-autonomous effects may not be easily detected with a WB) (Figure 3A-F). It would also be interesting to know the degree of overlap in up/down regulated genes between the experiments in Figure 2 and Figure 5, which was in contrast to Figure 2 performed with SST-Cre restricted KO of Nova1/2 (though, still with ECS).

While the effects of Kir2.1, NaChBac and either cKO or OE of Nova are all clearly cell autonomous, the effects ECS are, by definition, not. While the alignment of the NaChBac findings with the ECS results inclines us to believe they influence SST cINs in a similar manner, parsing the line between what aspects in ECS are induced directly versus indirectly is subtle and somewhat nebulous. While the reviewer suggests many further interesting areas, the need to focus was clear from both this reviewer and the others and we hope this reviewer concurs with the reasons for our choices. The work already spans an enormous range of approaches and has been expanded significantly over the past three years since its last review.

3. Following onto this previous comment, overall throughout the manuscript a more detailed and quantitative treatment of the alternative splicing results would be informative and lend greater credibility to the results presented here. While the authors provide a GO analysis, a coarse summary of alternative splicing "event types", and anecdotally mention a few example genes by name, fuller documentation of the exact genes that have been alternatively spliced would allow better vetting of the results. Furthermore, annotation of Nrxn1 is not correct: exon numbering oddly seems to go against strand direction and the coordinates appear to point at exon 6 (or splice site 2) rather than 10. Previously this exon has been shown to be uniformly excised in SST cells, which is the opposite of what this study finds. This should be clarified. Finally, although the authors have data to show this, there is no mention of how gene expression level changes after each manipulation (ECS, KOs…). It would be greatly reassuring if the authors clearly present which genes experienced the largest splicing changes and whether their expression level has changed significantly. Currently, it is impossible to tell, for example, if we are looking at splicing changes in genes, which expression has majorly gone down. While splicing changes occurred, gene expression may reveal more immediate insights into the phenotypic outcomes, axon and synaptic reorganization. This would be especially important in the case of ECS experiments, but also interesting when Nova is overexpressed.

We thank the reviewer for this suggestion. We have now included both GE and AS changes for all conditions tested. We find that although many synaptic genes have both GE and AS changes, the level of AS changes is larger than the changes in their GE. These data have now been included in both the main text and supplementary figures. Additionally, we have now included all the sashimi plots for the major synaptic genes involved in the github link which show that with ECS many genes are clearly AS, and those changes are largely abolished by knocking out Nova1/2. We analyzed the specific Nxrn1 exon that is differentially spiced in SST+ cINs in our data, we used the coordinates directly from the genome to label the exons and splice site.

4. In Figure 4, authors should quantify SST-IN axonal density in Nova 1 and Nova 2 cKOs in addition to the synaptic changes, as changes in axonal density were also observed in the activity manipulations of Figure 1. It would be also interesting to know if the cell changes in excitatory input onto SST-INs observed in Figure 5 with SST-restricted Nova1 or 2 KO are observed with activity manipulation as well. Although a lack of Nova2 KO impact on cell survival is casually mentioned in Discussion, this should be quantified and documented for each KO condition.

While we have examined excitatory input, our effort here was to narrow rather than broaden what was already a wide-reaching effort. We hope the reviewers concurs that by focusing on specifically efferent synaptic contacts of SST cINs, we are able to provide a more coherent story. Finally with regard to cell death, we explored this and note that this doesn’t appear to be affected and that the role of activity and synaptogenesis occurs during periods that are only weakly associated with the normal period of cell death.

5. Figure 2E is referenced in text, but not shown in figure.

This has now been corrected.

6. Without further clarification, I find the presentation of Nova1/2 localization unconvincing. Figure 3G shows that Nova1/2 is either cytoplasmic or nuclear or both. However, for quantification, apparently only the nuclear/cytoplasmic ratio was used. It should be clearly stated if "both" cells were counted as "nuclear" or not, because outcomes can be very different. Furthermore, it should be clarified if any quantitative measure or cutoff was used to determine cytoplasmic versus nuclear localization. Were these blinded experiments?

We used DAPI boundaries to locate the nucleus. We quantified the relative pixel intensity within the nuclear boundaries compared to outside to calculate the nuclear/cytoplasmic ratios.

[Editors' note: further revisions were suggested prior to acceptance, as described below.]

The manuscript has been improved but there are some remaining issues that need to be addressed, as outlined below:Please address each of the issues identified below. All of these changes should be able to be addressed with textual changes, and/or in one case a minor additional statistical analysis. The latter is concerned with Reviewer #2's point that results from 2/3 and 5 should be treated separately. In further consultation, the reviewers agreed that a simple test that the two are not different could be used to justify merging without further changes to figures or downstream analyses. It was also agreed that although a quantal measure from ChR2 experiments would be preferred because it affords better normalization, this issue could simply be acknowledged in the text.Reviewer #1 (Recommendations for the authors):The additional experiments go a large way toward addressing the key concerns raised by the reviewers. I have no further suggestions for improvement, other than two very minor clarifications.102 RNABPs (i.e. PTBP1/2, FUS, ELAVL4, SRRM4, Rbfox1, FMR1, Nova1, Nova2) → 102 RNABPs (e.g. PTBP1/2, FUS, ELAVL4, SRRM4, Rbfox1, FMR1, Nova1, Nova2)e.g. (as opposed to i.e.) makes it clear the listed RNABPs are examples."Sleepaway" should be replaced in the methods with a more generic description of the formulation or its components.

We thank the reviewer for pointing out these details. These have now been modified in the text.

Reviewer #2 (Recommendations for the authors):The authors performed a number of additional experiments to try to address the questions regarding the relationship between activity and Nova expression, which do improve the manuscript. It still is not crystal clear what the relationship is, but biology sometimes is not simple.Several remaining issues:The manuscript states that Nova proteins impact afferent and efferent connectivity through alternative splicing. I would say the manuscript falls a bit short of making that causal relationship. The data imply it, but no experiments were done specifically altering splice events individually or globally and assessing the impacts on connectivity.Authors report measuring mPSCs, however, no TTX was included in recording preparations. Therefore these would be better described as spontaneous PSCs, not minis.

We thank the reviewer for raising this point. These recordings were in fact miniature currents in the presence of TTX. We have now added this clarification in the methods section.

Why were results from L2/3 and L5 cells merged? They should also be assessed independently.A quantal measure of elicited PSC amplitude would still be a superior measure to the non-normalized version presented, even if the number of cells expressing ChR2 should theoretically not change.

As the reviewer suggested, we agree that a quantal measure would be superior, however the strontium experiments did not work well in our hands, and we were unable to elucidate quantal release. We believe using a transgenic Ai32 line can partially mitigate this issue.

The authors make the statement "we found a small decrease in the ratio of nucleus to cytoplasmic Nova protein within SST cINS injected with Kir2.1". It is unclear why this decrease is noted as small: the ratio (4.75 to.265) is approximately 20:1, similar to but larger than that for NaChBac (4.75 to 40.91, 1:10) listed as substantial. It is not clear why this wording was used.

The reviewer makes a valid point on the word choice here. We agree with the reviewer and have amended the wording to state that both Kir and NachBac induced significant changes in the nucleus to cytoplasmic ratio of Nova protein expression.

Differences in significance for GO results correlate with the number of altered AS genes identified. This perhaps is not a function of the biological significance of those genes identified in this way, but just the number of genes used as inputs.

That is indeed the case, it is the p-value associated with the number of genes annotated to be in a specific GO term, hence it does not reflect the biological significance or level of alternative splicing differences. However, we have attempted to check the significance of alternative splicing levels of synaptic genes under different experimental conditions as shown in Figures 2F, 4D, and 6F.

For S6F and S6H, it is unclear to me how a pval = 0.0001 could be obtained from these plots, at least with the values given in the text vs. the figure. Also, ** is noted, vs. *** which might be expected. I'm just confused by this. Is this also a difference in SEM vs. SD?

In Figure 5—figure supplement 1F right panel, the p-Value=0.005 as denoted in the figure legend is for the comparison of the average amplitudes under the different mutant conditions. However, in Figure 5—figure supplement 1H, since the cumulative fraction considers each event, by sheer total number of events the pValue tends to become really low making it p=0.0001. However, we realize this might be misleading, and we have now changed the pValue in the text to reflect the difference in the average amplitude as opposed to the cumulative fraction.